# Human pancreatic microenvironment promotes β-cell differentiation via non-canonical WNT5A/JNK and BMP signaling

Jolanta Chmielowiec[1,9], Wojciech J. Szlachcic [2,9], Diane Yang[1,9], Marissa A. Scavuzzo[3], Katrina Wamble[4], Alejandro Sarrion-Perdigones[5], Omaima M. Sabek [6,7], Koen J. T. Venken [5,8] & Malgorzata Borowiak [1,2,3,4,8 ✉]

In vitro derivation of pancreatic β-cells from human pluripotent stem cells holds promise as diabetes treatment. Despite recent progress, efforts to generate physiologically competent β-cells are still hindered by incomplete understanding of the microenvironment's role in β-cell development and maturation. Here, we analyze the human mesenchymal and endothelial primary cells from weeks 9-20 fetal pancreas and identify a time point-specific micro-environment that permits β-cell differentiation. Further, we uncover unique factors that guide in vitro development of endocrine progenitors, with WNT5A markedly improving human β-cell differentiation. WNT5A initially acts through the non-canonical (JNK/c-JUN) WNT signaling and cooperates with Gremlin1 to inhibit the BMP pathway during β-cell maturation. Interestingly, we also identify the endothelial-derived Endocan as a SST+ cell promoting factor. Overall, our study shows that the pancreatic microenvironment-derived factors can mimic in vivo conditions in an in vitro system to generate bona fide β-cells for translational applications.

[1] Molecular and Cellular Biology Department, Baylor College of Medicine, Houston, TX 77030, USA. [2] Institute of Molecular Biology and Biotechnology, Adam Mickiewicz University, ul. Uniwersytetu Poznanskiego 6, 61-614 Poznan, Poland. [3] Program in Developmental Biology, Baylor College of Medicine, Houston, TX 77030, USA. [4] Stem Cell and Regenerative Medicine Center, Baylor College of Medicine, Houston, TX 77030, USA. [5] Verna and Marrs McLean Department of Biochemistry and Molecular Biology, Baylor College of Medicine, Houston, TX 77030, USA. [6] Department of Surgery, The Methodist Hospital, Houston, TX, USA. [7] Weill Cornell Medical College, New York, NY, USA. [8] McNair Medical Institute, Baylor College of Medicine, Houston, TX, USA. [9] These authors contributed equally: Jolanta Chmielowiec, Wojciech J. Szlachcic, Diane Yang. ✉email: malbor3@amu.edu.pl

β-cells are the endocrine cell type of the pancreas and the only cells that produce and secrete insulin (INS) to maintain blood glucose homeostasis. Loss of functional β-cells is the central characteristic of diabetes, which can lead to retinopathy, neuropathy, and stroke, among other complications[1,2]. Currently available therapies require external insulin delivery that only approximately mimics physiological response to stabilize blood glucose levels[3–5]. The functional cure for diabetes would require transplantation of β-cells as a replacement for lost or dysfunctional β-cells[6]. In fact, cadaveric islets can restore normoglycemia in humans[7]. Unfortunately, each transplant requires billions of cells, and due to scarcity of cadaveric islets, there has been enormous interest in creating a new and robust source of β-cells to treat diabetes. β-like cells derived from in vitro differentiation of human pluripotent stem cells (hPSCs) can alleviate hyperglycemia in mice[8–14]. Although tremendous progress has been made recently, hPSCs cannot yet be reliably coaxed into functional β-cells in sufficient purity, numbers and costs to serve for therapy in the clinics.

The directed pancreatic differentiation of hPSCs commonly aims to mimic in vivo development in an in vitro system. Yet, in vitro systems are often simplified and lack the components of the microenvironment present in vivo. The microenvironment is a complex, multicellular system, composed of mesenchymal, endothelial, neuronal, and immune cells, and structural molecules, which are part of the extracellular matrix (ECM)[15–21]. Multiple recent studies, including single-cell RNA-seq experiments, highlighted the sophisticated cellular diversity of the microenvironment that is organ- and time-specific. This dynamic complexity reflects various interactions between different cell types in the developing organs and this crosstalk is critical for proper organ development and function.

In pancreas, the microenvironment provides mechanical and chemical cues that are indispensable for the initial fate commitment, developmental growth, spatial organization and functional maturation of endocrine and exocrine cells[21–24]. Multiple signaling pathways controlling pancreatic epithelium development regulated by microenvironment-released factors were identified, including FGFR2, retinoic acid, BMP, TGFβ, Notch, Hedgehog and canonical Wnt pathways[25–34]. This knowledge has been adapted for human β-cell in vitro differentiation, enabling great advancements in pancreatic cell derivation. However, inclusion of the cellular microenvironment components, such as mesenchymal and endothelial (M-E) cells, at various stages of pancreatic differentiation is still beneficial for the development of mouse and human pancreatic β-cells and their progenitors[35–40]. This indicates that further identification of signals and molecular mechanisms beyond microenvironment-epithelium crosstalk is crucial for better understanding and controlling β-cell derivation.

Recently, we and others showed that mesenchyme in developing mouse pancreas dynamically changes over time[41,42]. This likely reflects the prerequisite to adjust M-E released signals to drive developmental progression at a correct pace, as the same signaling pathway can antagonistically influence consecutive developmental stages. However, the knowledge of factors released by human M-E remain incomplete, together with the underlying mechanisms that guide pancreatic endocrine progenitors (EP) decisions to become a specific endocrine cell type. For example, endocrine cells are formed in timely waves during human pancreas development—the first wave β-cells at weeks 8–9, followed by α-cells that express glucagon (GCG) at weeks 14–16, and finally by the second wave of β-cells at weeks 17–21, which accounts for the majority of adult β-cells[43–45]—but the regulatory mechanisms beyond remain unknown.

In this work, we hypothesize that: (1) interactions between human EPs and the pancreatic M-E cells promote the differentiation and maturation of these progenitors into β-cells, and (2) components of the human pancreatic microenvironment change over time and differ in their capability to promote β-cell specification. We, therefore, use a two-pronged strategy and first, we establish an in vitro model of a primary human fetal pancreatic microenvironment, composed of M-E cells at different developmental stages, to delineate their contribution to differentiating human β-cells. After determination which stages stimulate β-cell development the most, we identify the M-E-derived signals that promote the specification of endocrine fate. We further characterize the molecular events associated with in vitro β-cell differentiation driven by the identified growth factors.

## Results

**Human pancreatic niche-derived cells β-cell fate**. To understand how the components of pancreatic niche, specifically mesenchyme and endothelium at different time points, affect endocrine differentiation, we obtained human pancreas and other endodermal organs at fetal week 9.1, 10.6, 13, 14.6, 16.3 (separated as body and head of the pancreas), 17.5 (separated as body and head of the pancreas), and 20.1. These time points reflect a window when most EPs and endocrine cells form in the human pancreas[43–46]. We then derived 12 stage- and organ-specific human M-E primary cells (Fig. 1a) that we named M-E9 through 20. For each time point, we did 2–3 independent derivations from different organ dissections. First, we extensively characterized the de novo established M-E primary cells using immunofluorescence and confirmed expression of mesenchymal (VIM)[47] and endothelial markers (PECAM1, CFIII)[48] in these cells (Fig. 1b). We also used qRT-PCR to quantify mRNA levels of mesenchymal (VIM, FSP1)[47,49] and endothelial markers (PECAM1, FLK1, CDH5, ICAM, and VWF)[48,50–53] in the M-E cells (Supplementary Fig. 1A, B). All tested markers were expressed at levels comparable to their respective controls, i.e., mesenchyme-like human neonatal dermal fibroblasts (HDFs), human endothelial cells (umbilical vein endothelial cells, HUVECs)[54], and murine pancreatic endothelial cells (Mile Sven1, MS1)[55], illustrating their M-E origin. The M-E cells maintained the expression of mesenchymal markers for at least 16 passages, however, the endothelial gene expression decreased after 8 passages (Supplementary Fig. 1C). Therefore, to keep the composition of M-E cells closer to the in vivo niche, we performed all subsequent experiments with M-E cells within the first 6 passages. Further, we did not detect expression of any epithelial pancreatic progenitor (PP) markers, such as PDX1 or SOX9[56,57] and β-cell marker INS in any of the primary cell cultures (Supplementary Fig. 1D–F).

To determine how the different M-E cells influence endocrine cell differentiation, specifically whether they promote multipotent PP differentiation into β-cells, we cocultured each of the M-E primary cells with hESC-derived PPs. The hESCs (HUES8) were guided in a stepwise manner toward the pancreatic fate using the modified published protocol[35] (Supplementary Fig. 2). At the PP stage, the protocol consistently yielded 60–80% cells positive for PDX1 and SOX9 (Supplementary Fig. 2), which jointly drive PPs development[58]. PDX1 was also coexpressed in around 50% of PPs with NKX6-1, a downstream pro-endocrine transcription factor[59] (Supplementary Fig. 2). At the EP stage, ~18% cells coexpressed endocrine cell markers: transient master regulator NGN3[46,60,61], and pan-endocrine marker Chromogranin A (CHGA)[62] (Supplementary Fig. 2) while ~40% cells expressed CHGA only.

The hESC-derived PPs were replated and cocultured with each of 12 M-E primary cell types in a basal medium containing no β-cell inducing factors (Fig. 1c). To determine the specificity of M-E cells, we also cocultured PPs with HUVECs, HDFs, and mouse embryonic fibroblasts (MEFs), none of which are present in the

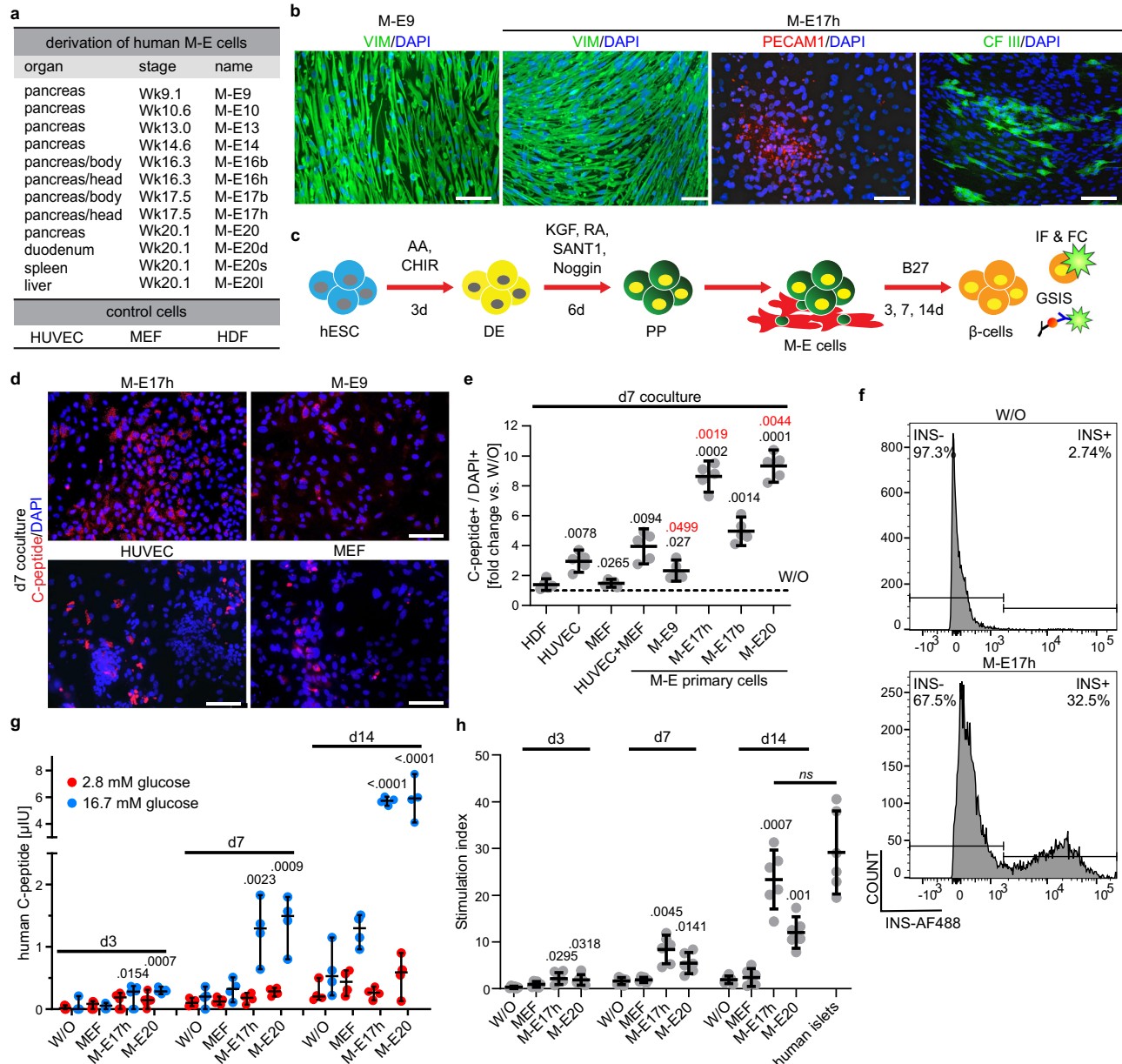

**Fig. 1 Human β-cells generated in coculture with organ- and stage-specific mesenchymal-epithelial (M-E) cells secrete INS in response to high glucose levels. a** Human M-E primary cells were derived from different organs and developmental stages. Primary mesenchymal cells (MEFs, HDFs) and an endothelial cell line (HUVECs) were controls. **b** Immunofluorescent staining of primary cells for mesenchymal VIM or endothelial markers, PECAM1 and CF III. DAPI marks nuclei. Scale bar = 100 μm. See also Supplementary Fig. 1. N = 3 independent experiments. **c** Coculture approach to induce β-cells from PPs. hESCs-derived PPs were cultured on M-E cells for 3, 7 or 14 days. At these time points, cells were analyzed by immunofluorescence combined with microscopy (IF) or flow cytometry (FC), or glucose-stimulated insulin secretion (GSIS). **d** β-cells probed with C-peptide antibody after 7-day coculture with M-E17, M-E9, HUVECs, or MEFs. DAPI, marks nuclei. Scale bar = 100 μm. N = 5 independent experiments. **e** Scatter plot representing IF quantification of C-peptide+ cells from cocultures presented as fold change normalized to without (W/O) coculture (dashed line). Mean ± 95% CI and significant *p* values (Dunnett's multiple comparisons test) vs. W/O or HUVECs + MEFs (in red) are shown. N = 5 independent experiments. **f** Flow cytometry plots of INS+ cells at day 7 of PPs without coculture (left) or cocultured with M-E17 cells (right). **g** GSIS after PP coculture with M-E cells. Scatter plot represents ELISA measurements of human C-peptide secreted after 2.8 mM and 16.7 mM glucose treatment at day 3, 7, or 14 of coculture with M-E and control cells. N = 6 (day 3) or 4 (day 7, 14) independent experiments. Results were normalized to cell number and total protein content. Mean ± 95% CI, and significant *p* values (Dunnett's multiple comparison test) vs. W/O are shown. See also Supplementary Fig. 9b–d. **h** Scatter plot with data points representing stimulation index for coculture at day 3, 7, and 14, calculated as a ratio of insulin secreted after 16.7 to 2.8 mM glucose. N = 6 independent experiments. Mean ± 95% CI and significant *p* values (Dunnett's multiple comparison test) vs. W/O are shown. ns indicates no statistical difference.

developing pancreas in vivo. After 7 days of the coculture, cells were stained for C-peptide, a byproduct of INS production, and imaged using confocal microscopy (Fig. 1d). Coculturing PPs for 7 days with M-E17 or M-E20 cells increased the number of C-peptide+ cells by 8.6–9.3- or 2.2–2.4-fold, compared to no

coculture (W/O) or coculture with HUVECs + MEFs, respectively (Fig. 1e). Interestingly, coculturing PPs with M-E9 increased the number of C-peptide+ cells only by 2.3-fold over W/O condition, similarly to HUVECs (Fig. 1e). We also used flow cytometry to quantify INS+ cells after 7-day coculture of PPs

with M-E17 and found 30% of live cells were INS+ compared to 3% in the control (Fig. 1f). Finally, we confirmed that INS+ cells induced in the presence of stage-specific M-E cells co-express NKX6-1, a marker of monohormonal β-cells (Supplementary Fig. 3A). We did not see any further increase in C-peptide+ cell induction after extending the coculture time to 14 days, suggesting that M-E cells mostly stimulate endocrine fate acquisition of existing progenitors rather than increasing C-peptide+ cell proliferation.

**M-E-induced human β-cells are glucose-responsive in vitro.** Glucose-stimulated INS secretion (GSIS) is the critical physiological assessment of hPSC-in vitro-derived β-cells. We performed GSIS after coculturing progenitors with M-E9, M-E17, and M-E20 cells for 3, 7, and 14 days and quantified the secreted human C-peptide. We first assessed the ability of INS+ cells derived in coculture with M-E9 and M-E20 for 3, 7, and 14 days to secrete INS and C-peptide. For this, we stimulated β-cells with 30 mM KCl, which depolarizes the cell membrane causing secretion of INS and C-peptide from vesicles docked at the membrane. Quantification of C-peptide after 3 days of coculture showed secretion of C-peptide (1 μIU for M-E17 coculture and 2 μIU for M-E20) in contrast to control (Supplementary Fig. 3B). Significantly more C-peptide upon KCl stimulation was released after longer coculture with M-E17 (7.5 μIU and 12.8 μIU at day 7 and 14, respectively) (Supplementary Fig. 3C, D). These results suggest enhanced differentiation of PPs cocultured with ME into INS-producing cells.

We then asked the question of whether our hPSC- derived β-cells exhibit GSIS in any of the given time points. Contrary to KCl-induced INS/C-peptide release, GSIS is a functional test for β-cells. We performed a GSIS assay at basal blood glucose levels (2.8 mM), and at elevated blood glucose levels (16.7 mM) that innately induce secretion of INS along with C-peptide. Although PPs cocultured with M-E for 3 days released small amounts of C-peptide after KCl stimulation, they were not glucose-responsive (Fig. 1g and Supplementary Fig. 3B). Conversely, after a longer coculture time, cells acquired responsiveness to the high glucose concentration, reaching the highest GSIS after 14 days (Fig. 1g and Supplementary Fig. 3B–D). The PPs cultured with M-E17 had the greatest stimulation index, defined as a ratio of high-to-low glucose-induced C-peptide secretion, of 23.4, compared to 1.9 and 2.4 in W/O and MEF coculture, respectively (Fig. 1h). The same GSIS tests on six independent batches of human cadaveric islets from healthy donors yielded a stimulation index of 29.2, not significantly different from that of PPs cocultured with M-E17 (Fig. 1h). Together, these data show that β-cells derived in the presence of stage-specific primary pancreatic M-E cells are glucose-responsive, reaching GSIS comparable to that of human cadaveric islets.

**Growth factors and ECM from M-E induce β-cell fate in vitro.** Secreted factors, cell-cell interactions, and ECM interactions mediate the crosstalk between the pancreatic niche and progenitors. To dissect whether M-E cells can promote β-cell differentiation without cell-cell contacts permitted in our coculture experiments, we designed PP culture on M-E-derived ECMs, and separately in conditioned media (CM) obtained from M-E culture in serum-free media (Fig. 2a). To evaluate the ECM effect, we decellularized M-E9, M-E17 and M-E20 cultures using non-enzymatic methods. The cell-depleted plates were positively stained for Collagen IV and Laminin, known ECM components (Supplementary Fig. 4)[63]. We then cultured PPs on the different ECMs for 14 days (Fig. 2b, c). Further, we determine the impact of soluble secreted factors alone, by culturing PP cells in CM (Fig. 2d, e). As a negative control, we used heat-inactivated CM and

found almost no C-peptide+ cells, similarly to the culture without CM. C-peptide staining showed that both ECM and CM promoted β-cell differentiation (Fig. 2c, e). However, the efficiencies were lower than in the coculture experiments (see Fig. 1e), pointing to complexity of M-E mediated pro-endocrine function.

Despite the higher efficacy of β-cell differentiation in cocultures, this method has a limited potential for regenerative medicine. Knowing that CM from specific M-E cells significantly promotes PP maturation into INS+ cells, we focused on defining the M-E secreted factors, which can be easily applied into the culture medium during in vitro differentiation.

**Identification of secreted growth factors from M-E cells.** During pancreatic differentiation, PPs must traverse through the EP stage to become β-cells. As it takes 4–5 days to induce EPs from PPs in vitro, we tested the influence of M-E cells on EPs in a 3-day coculture (Fig. 2f). We found that 3-day EPs coculture with M-E17 or M-E20 led to an 9.3 and 10.5-fold, respectively, in β-cell formation (Fig. 2g) and a significant improvement in GSIS by 13.3 and 12.5-fold for M-E17 and M-E20, respectively (Fig. 2h). These results show that the M-E derived signals involved in β-cell specification and maturation are received and processed by EPs.

To identify the M-E-derived signals, we transcriptionally profiled M-E cells by RNA-sequencing (RNA-seq). Principal component analysis of the RNA-seq data revealed that M-E9, M-E20, and M-E17 clustered apart from HUVEC and HDF controls, indicating that the differences between the M-E cells and controls are more defined than those between the M-E cell types (Fig. 3a). Functional annotation of the upregulated genes shared by M-E17 and M-E20 showed the enrichment in glycoproteins, secreted proteins, ECM components, and signaling pathway-associated genes (Fig. 3b), categories that may play a role in PP and microenvironment crosstalk. Within the signaling gene ontology terms, the Integrin, Wnt and Cadherin pathways were among the most enriched pathways in M-E17 and 20 cells (Fig. 3c). Also, genes encoding secreted factors and proteins involved in signaling such as *WNT5A*, *SERPINF1*, *HGF*, *LIF, PDPN, UCN2*, and *DCN* were upregulated in M-E17 or M-E20 (Fig. 3d).

To determine EP receptivity to M-E signals we also profiled hPSC-derived EPs by RNA-seq. For this, we cross-referenced our RNA-seq data with the FANTOM5 database[64] to perform ligand-receptor connectome analysis (Fig. 3e, f). Connectome analysis revealed ligands from M-E9, M-E17, and M-E20 cells that potentially bind to the receptors expressed in hESC-derived EPs (Fig. 3e), highlighting M-E derived ligands that may lead to the increase in INS+ cells after coculture. We found that out of the candidate factors, the most M-E ligand-EPs' receptor pairings were of ligands derived from M-E17 and M-E20 (22 ligands each, Fig. 3f). Importantly, some ligands uncovered in differential expression analysis of M-E cells, including *WNT5A*, *LIF*, *HGF*, and *THBS2*, had their receptors expressed in EPs, suggesting they can affect EP differentiation. However, this analysis depends on current knowledge regarding ligand-receptor pairing and their high expression level, and the importance of other ligand-receptor pairs cannot be excluded. For instance, *ESM1* (also known as Endocan), *SERPINF1*, *PDPN*, and *WNT5B* were not covered by the FANTOM database. Among identified genes, some encode already known factors promoting β-cell differentiation in vitro, including FGF7 (KGF) or EGF[8,65,66]. We set out to investigate factors with unknown contribution to human pancreatic endocrine differentiation.

**WNT5A efficiently induces EP differentiation into INS+ cells.** By comparing genes significantly enriched in both M-E17 and M-E20 cells to those that interact with receptors expressed on

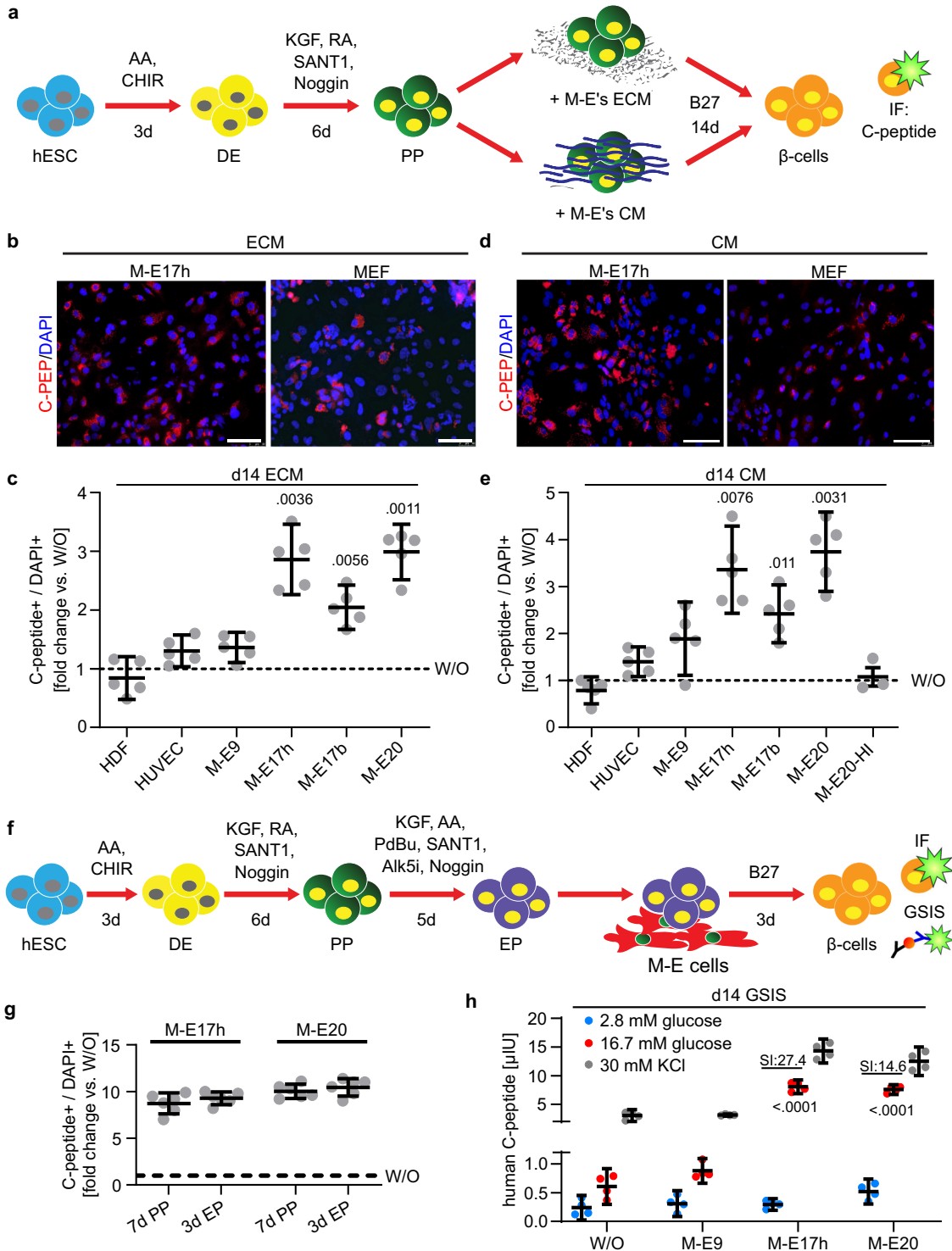

hESC-derived EPs, we selected EGF, Endocan, FGF7, HGF, IGF1, LIF, PDPN, SERPINF1, THBS2, WNT3A, WNT5A and WNT5B proteins (Fig. 4a) for further studies. We first confirmed their expression in M-E17 and M-E20 cells using qRT-PCR (Supplementary Fig. 5). Then, we asked if any single growth factor would increase the endocrine cell number after 3-day treatment of EPs (Fig. 4b), which we assessed by IF for the endocrine marker—CHGA (Fig. 4c, d). Except for LIF and WNT3A, all tested proteins significantly increased the fraction of CHGA+ cells compared to untreated control in at least one concentration used (Fig. 4d). The most prominent effect on CHGA+ cell number

had a treatment with SERPINF1 (2.2-fold over control at 1 µg/ml concentration), WNT5A (2.5-fold, 500 ng/ml) and Endocan (2.5-fold, 50 ng/ml). We also monitored Islet-1 (ISL1) expression using a ISL1$^{Cre/+}$; pCAG$^{loxP-STOP-loxP-EGFP}$ hESC line engineered for lineage tracing of ISL1+ cells[67]. Factors that significantly enriched CHGA+ cells show similar induction of ISL+ endocrine cells as assessed by GFP fluorescence (Supplementary Fig. 6A, B).

To test β-cell specification by single factors, we analyzed INS and C-peptide protein presence after a 3-day EPs treatment (Fig. 4e, f and Supplementary Fig. 6C, respectively). EPs treatment with PDPN, SERPINF1, WNT5A and WNT5B

**Fig. 2 ECM or conditioned media from M-E cells increase C-peptide positive cells in pancreatic progenitors. a** Overview of approach to determine the contribution of the extracellular matrix (ECM) or conditioned media (CM) from M-E cells to PP differentiation into β-cells. **b** Images of PPs cultured for 14 days on the ECM derived from M-E or MEF cells and probed with anti-C-peptide antibody. DAPI marks nuclei. Scale bar = 100 μm. N = 5 independent experiments. **c** Scatter plots of C-peptide+ cell quantification after 14-day culture of PPs with ECM as a fold change over basic media (W/O). Mean ± 95% CI, and significant p values (Dunnett's multiple comparison test) vs. W/O are shown. N = 5 independent experiments. **d** Images of PPs cultured for 14 days in conditioned media (CM) derived from M-E or MEF cells and probed with anti-C-peptide antibody. DAPI marks nuclei. Scale bar = 100 μm. N = 5 independent experiments. **e** Scatter plots of C-peptide+ cells quantification after 14-day culture of PPs with conditioned media (CM) as a fold change over basic media (W/O). Mean ± 95% CI, and significant p values (Dunnett's multiple comparison test) vs. W/O are shown. M-E20-HI is a heat-inactivated CM from M-E20 cells. N = 5 or 3 (HI control) independent experiments. **f** Approach to determine if EP coculture with M-E17 or M-E20 primary M-E cells potentiates C-peptide+ cells. **g** Quantification of IF data showing C-peptide+ cells after either PP or EP coculture with M-E cells for 7 or 3 days, respectively. Mean ± 95% CI fold change normalized to non-coculture control (W/O) is shown. N = 6 independent experiments. **h** GSIS analysis at day 3 of EP coculture with M-E cells. Scatter plot representing ELISA measurements of human C-peptide after stimulating with 2.8 mM and 16.7 mM glucose. N = 4 independent experiments. Results were normalized to cell number and total protein content. Mean ± 95% CI, and significant p values (Dunnett's multiple comparison test) vs. W/O control are shown. Additionally, stimulation index (SI) values, counted as in Fig. 1h, are shown for the effective M-E17 and M-E20 conditions.

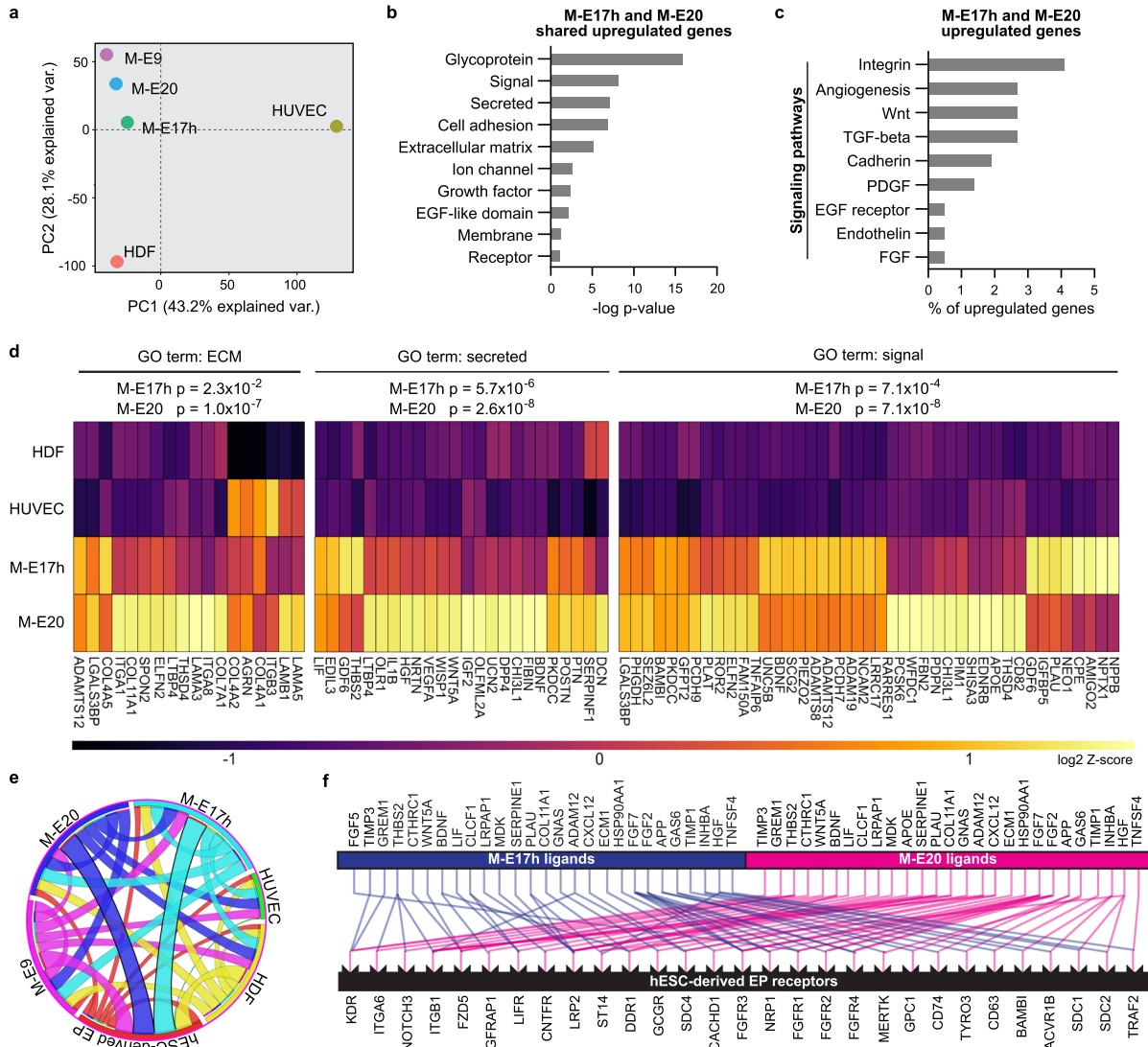

**Fig. 3 Human pancreatic M-E17 and M-E20 cell transcriptomes are jointly enriched in secreted factors that have their receptors present in hESC-derived EPs. a** Principal component analysis of human primary M-E cells and control cell transcriptomes showing that pancreatic M-E cells are distinct from other HDFs and HUVECs. **b** Functional gene annotation of significantly enriched genes upregulated in both M-E17 and M-E20 (fold change >20). **c** Signaling pathways represented by unique genes significantly upregulated in both M-E17 and M-E20 (fold change >20). **d** Heatmap of normalized $\log_2$ Z-score expression of ECM, secreted growth factors and signal proteins enriched in human M-E17 and M-E20 cells compared to HDFs and HUVECs. Fisher's exact test was used to estimate p values. **e** Ligand-receptor connectome analysis based on data from FANTOM5 database and RNA-Seq. Lines connect ligands expressed by M-E cells, control cells, with their receptors expressed in hESC-PPs. Lines' thickness correlates with the number of such ligand-receptor pairs between cell types. **f** Graph presenting ligands expressed by M-E cells that have their known receptors expressed in hESC-derived PPs.

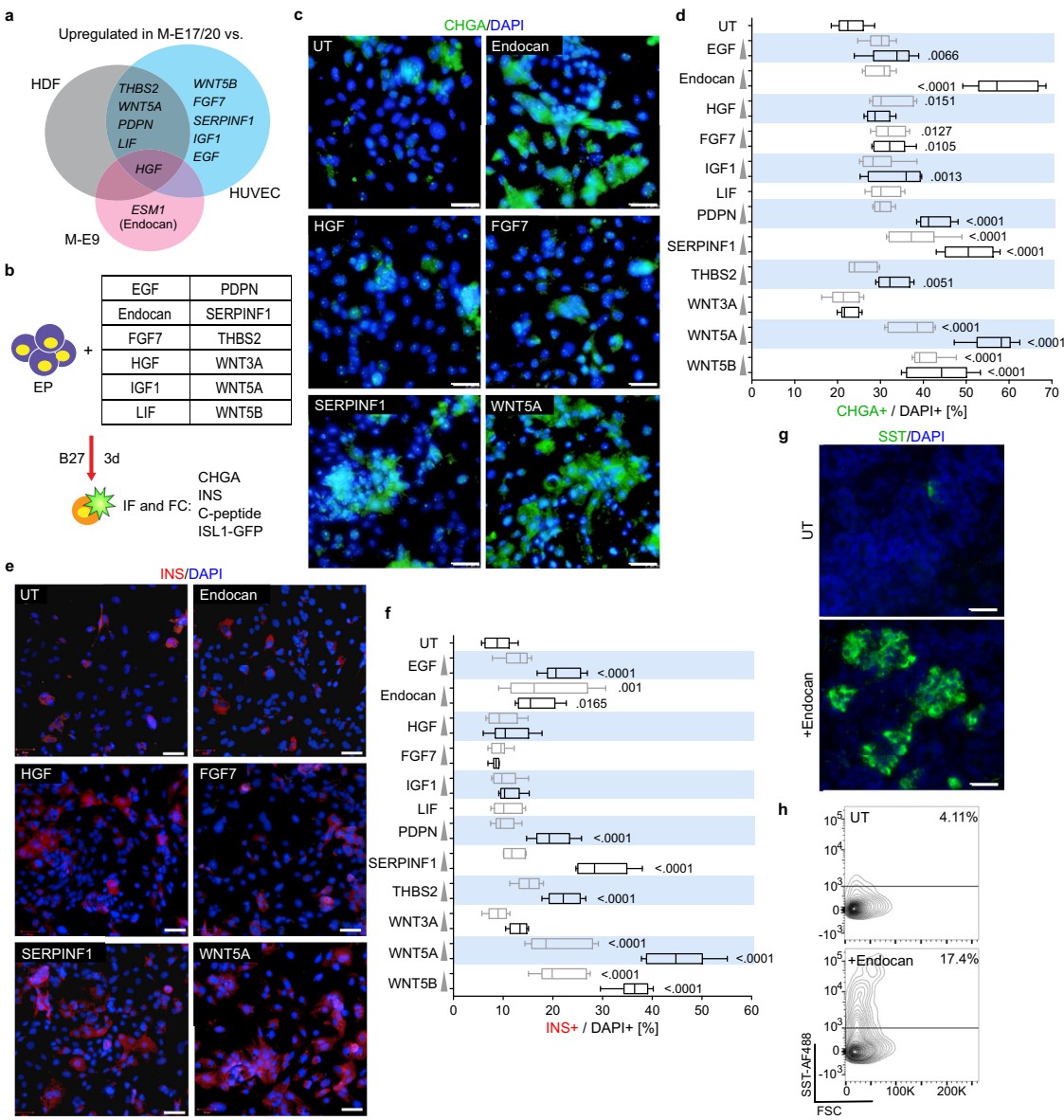

**Fig. 4 Selected growth factors secreted from M-E cells differentiate hESC-derived endocrine progenitors into CHGA- and INS-positive cells. a** Venn diagram presenting growth factors selected for further testing and their upregulation in M-E17 and M-E20 compared to M-E9, HDFs and HUVECs. **b** Experimental design, where hESCs were differentiated into endocrine progenitors (EPs) and then incubated with basal media only or with individual M-E derived growth factors at two concentrations (Supplementary Table 3) for 3 days, followed by immunofluorescence analysis. See also Supplementary Fig. 4. **c** Images of CHGA (green) immunofluorescent (IF) stainings of EPs cultured in basal media only (UT) or in the presence of growth factors. DAPI marks nuclei. Images of high concentration treatment (Supplementary Table 3) are shown. Scale bar = 100 μm. N = 6 independent experiments. **d** IF quantification of CHGA+ cells after treatment of EPs with growth factors, shown as % out of total (DAPI+) cells. Boxes extend from 25th to 75th percentile, middle lines denote median, while whiskers show minimum and maximum values. N = 6 independent experiments. p values (Dunnett's multiple comparisons test) are shown for conditions significantly different from the untreated (UT) control. **e** Images of INS F stainings of EPs cultured in media only (UT) or in the presence of example growth factors. DAPI marks nuclei. Images of high concentration treatment (Supplementary Table 3) are shown. Scale bar = 100 μm. N = 6 independent experiments. **f** Quantification of INS+ cells after treatment of EPs with growth factors, shown as % out of total (DAPI+) cells based on IF staining. Boxes extend from 25th to 75th percentile, middle lines denote median, while whiskers show minimum and maximum values. N = 6 independent experiments p values (Dunnett's multiple comparisons test) are shown for conditions significantly different from the untreated (UT) control. **g** Images of SST IF stainings of EPs cultured in media only (UT) or in the presence of Endocan. Scale bar = 100 μm. N = 3 independent experiments. **h** Flow cytometry plot with quantification of SST+ cells in EPs cultured in media only (UT) or in the presence of Endocan. N = 3 independent experiments.

increased numbers of both INS+ and C-peptide+ cells by 2- to 5-fold, compared to untreated controls (Fig. 4e, f and Supplementary Fig. 6C, respectively). This efficacy was confirmed with EPs derived from another hESC line, H1 (Supplementary Fig. 6D). Out of tested factors, WNT5A had the most potent, dose-dependent effect, at 500 ng/ml causing on average, a five-fold increase in the number of β-cells.

We were intrigued by the effects of Endocan treatment, known as an endothelial-derived factor[68]. Although this factor induced CHGA+ cells as efficiently as WNT5A, WNT5B and SERPINF1, the effect was not reflected by INS+/C-peptide+ cell induction. We, therefore, sought to identify the endocrine cell type induced by Endocan. Excitingly, we found more than a four-fold increase in SST+ cells after Endocan treatment compared to controls

(Fig. 4g, h), whereas GCG+ α-like cell number was not changed (Supplementary Fig. 6E). Therefore, we identified an M-E cell-derived factor, Endocan, as the specific inducer of pancreatic δ-cell fate. To our knowledge, Endocan is the first growth factor identified as an inducer of SST expression in human ESC-derived EPs.

Further, we investigated whether M-E derived factors cooperate to promote the differentiation of human endocrine cells. We selected 3 factors that most significantly promoted endocrine differentiation in our experiments (Endocan, SERPINF1, WNT5A) and HGF, based on literature search, and tested their effect on hESC-derived EPs in combinations of two, three or four growth factors (Supplementary Table 4). After 3 days of combinatorial treatment of EPs, we evaluated the number of INS+ cells (Supplementary Fig. 7A–C). For almost all combinations INS+ level was increased similarly to single growth factor treatment, and (Supplementary Fig. 7A, B) only combined treatment with Endocan and HGF outperformed each of them alone, with a 1.8-fold increase over Endocan alone and a 2.6-fold increase over HGF treatment. Still, the Endocan + HGF treatment was less effective than WNT5A alone. Also, HGF showed an inhibitory effect in combination with WNT5A, as fewer INS+ cells were induced than with WNT5A alone. Flow cytometry confirmed the lack of synergy between WNT5A and other factors, as compared to WNT5A alone (Supplementary Fig. 7C, D), further supporting the leading contribution of WNT5A.

As the next step, we tested whether WNT5A promotes β-cell formation in vitro using different pancreatic differentiation protocols that vary in culture dimensionality, growth factors and basal media composition. We used three additional protocols[9,10,69] to differentiate hESCs toward EPs (Supplementary Fig. 7E). In two of these protocols, we transferred PPs into a suspension, 3D spheroid culture, mimicking the 3D nature of a developing pancreas. At the EP stage of each protocol, we applied 500 ng/ml of WNT5A. After a 4-day-treatment we observed 1.5 to 2.2-fold induction of INS+ cells compared to controls where only protocol-specific growth factors were used (Supplementary Fig. 7E). These data support our finding that WNT5A effectively promotes β-cell formation using various pancreatic differentiation protocols.

Together, these experiments identified a M-E derived WNT5A, as a potent inducer of INS+ cells from human EPs. Therefore, we set out to study the molecular mechanism of WNT5A-mediated β-cell differentiation.

**WNT5A is sufficient and necessary to induce human β-cells.** First, we confirmed WNT5A expression in the pancreatic niche in vivo using immunostaining. WNT5A is expressed in VIM+ (i.e., mesenchymal) and PECAM1+ (i.e., epithelial) cells in week 16 pancreas (Fig. 5a, b). Next, we set up complementary experiments to recombinant WNT5A protein supplementation. We transiently overexpressed WNT5A in hESC-derived EPs and observed a dose-dependent increase in the number of WNT5A+ cells within 3 days after transfection with pCDNA-WNT5A plasmid (Supplementary Fig. 8A). The higher dose of the pCDNA-WNT5A plasmid induced a 150% increase in the number of WNT5A expressing cells (Supplementary Fig. 8A) followed by a 16-fold increase in the number of INS+ cells (Fig. 5c, d) as compared to transfection with the backbone plasmid. Next, we treated EPs with 1 μg of anti-WNT5A antibodies to block WNT5A signaling[70,71] and detected a 2-fold decrease in the number of INS+ cells (Supplementary Fig. 8B), further showing that WNT5A signaling promotes in vitro human β-cell differentiation.

To test WNT5A necessity for human β-cell differentiation, we disrupted *WNT5A* expression in M-E17 and M-E20 cells by

targeting the first constitutive exon (exon 3) using CRISPR-Cas9 nickase (W5A KO) and two independent pairs of sgRNAs (1 + 2) and (3 + 4) (Supplementary Fig. 8C). A neomycin-resistance cassette was inserted into exon 3 to introduce the frameshift and to select the positive clones before knockout confirmation by external and internal PCRs (Supplementary Fig. 8D), followed by IF (Fig. 5e)[72]. Coculture of EPs with M-E20 W5A KO cells yielded nine-fold fewer INS+ cells than coculture with control (WT) M-E20 cells (Fig. 5f). Further, by adding WNT5A to the coculture of EPs and W5A KO, we partially rescued the INS+ cell induction (Fig. 5f). We also cocultured EPs with M-E20 W5A KO cells in 3D format and again observed a profound decrease in INS + cells (Fig. 5g). Finally, we observed a similar decrease in coculture of M-E17 W5A KO cells with EPs (Fig. 5g). Together, these results appoint WNT5A as a crucial β-cell promoting factor released by pancreatic M-E cells.

**WNT5A coaxes β-cells by non-canonical WNT and JNK signaling.** We next investigated molecular mechanisms of WNT5A responsible for EP transition to β-cells. We noted that the prolonged (12-day) treatment with WNT5A of EP-stage spheres does not result in more INS+ cells, as compared to control based on Pagliuca et al. protocol (Supplementary Fig. 9A, B), suggesting that WNT5A accelerates β-cell differentiation of the existing EPs pool. Of note, earlier (Fig. 4 and Supplementary Figs. 6 and 7) we tested the ability of WNT5A to induce INS expression in the absence of other growth factors, while in the spheroid experiment (Supplementary Fig. 9A, B) we added WNT5A in combination with T3 and ALK5i included in the Pagliuca et al. protocol. To confirm that WNT5A does not act through proliferation, we used the mitotic marker phospho-histone 3 (pH3) and 5-ethynyl-2′-deoxyuridine (EdU) incorporation assays (Supplementary Fig. 9C, D). The WNT5A treatment increased the pH3+ cell number from 1 to 2% of total cell, while no significant change was observed in the EdU incorporation assay, indicating WNT5A has only a minor effect on proliferation. Since WNT5A treatment increases the number of INS+ cells by five-fold, it suggests that WNT5A acts through differentiation rather than proliferation.

WNT5A can activate both non-canonical and canonical WNT pathways[73,74]. To investigate which pathway is activated in WNT5A-induced β-cell formation, we tested the canonical β-catenin-dependent pathway in WTN5A-treated EPs using the TOPflash reporter[75]. EPs were transfected with either TOPflash or FOPflash and treated with WNT5A or a GSK3-β inhibitor CHIR99021 as a positive control. Untreated EPs had low TOPflash activity, and WNT5A treatment did not significantly activate or antagonize the β-catenin-dependent pathway (Supplementary Fig. 10A). Therefore, we hypothesized that WNT5A activates the non-canonical pathway in EPs, which regulates calcium signaling, planar cell polarity (PCP) and migration[76]. RNA-Seq showed that FZD3, a non-canonical WNT receptor activated by WNT5A[77], is expressed in human EPs, and we detected FZD3 protein in hESC-derived INSM1 + EPs[78] (Fig. 6a). Next, we treated hESC-derived EPs with WNT5A and FZD3-neutralizing antibodies and observed a 2.5-fold decrease in INS+ cell number compared to WNT5A treated EPs (Fig. 6b). These results suggest that FZD3 expressed in EPs plays a role in WNT5A-mediated INS upregulation.

To investigate the downstream targets of WNT5A in EPs, we performed RNA-seq of cells treated with WNT5A over short-term (12 h) and long-term (5 days) (Fig. 6c). As expected, the 5-day WNT5A treatment resulted in pronounced upregulation of β-cell secretory genes and transcription factors as well as glucose processing and membrane channels genes, suggesting ongoing β-cell maturation (Fig. 6d). For instance, WNT5A lead to an 81-fold

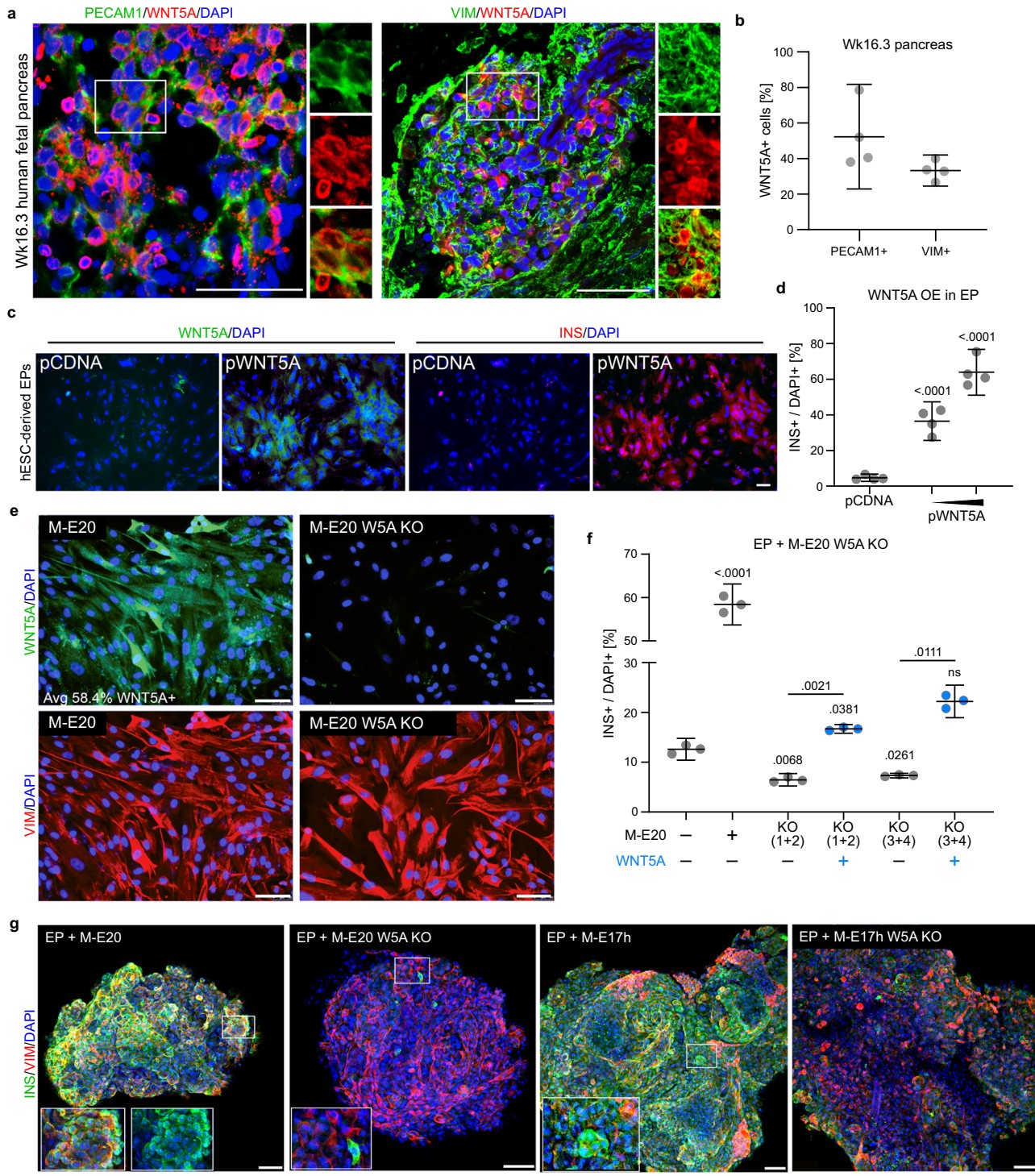

increase in *INS* and 56-fold increase in the transcription factor *NEUROD1*, as well as increases in glucose processing and insulin-secretion regulators including *GCK* (9-fold), *PCSK2* (7-fold) and *SYT4* (16-fold). Consistently with WNT5A specifically promoting β-cell differentiation, we also found downregulation of genes encoding EP markers, including *NGN3* (2-fold), or mature α-cell markers: *MAFB* (3-fold), and *GCG* (5-fold). We showed increased expression of *INS*, *CHGA*, *ONECUT1* and *PCSK2* but decreased of *GCG* in 5-day WNT5A-treated cells, verified by qRT-PCR (Supplementary Fig. 10B).

Given the strong pro-β-cell transcriptomic profile induced by the long treatment of WNT5A, we asked which pathways are

activated after the short treatment with WNT5A in efforts to uncover WNT5A effectors in hEPs. Gene Set Enrichment Analysis (GSEA) determined a gene set associated with the JUN kinase (JNK) pathway (Fig. 6e) and numerous genes regulated by the JNK pathway were significantly upregulated by 12-h WNT5A treatment (Fig. 6f). We also performed pathway analysis using TFactS[79] to predict regulation of known transcription factors based on upregulated and downregulated genes after 12-h of WNT5A treatment. The TfactS analysis identified JUN as the most significantly regulated transcription factor (Fig. 6g). The JNK-mediated phosphorylation regulates activity of JUN transcription factors. Short-term WNT5A treatment caused increased

**Fig. 5 WNT5A is expressed in the human pancreatic niche during development and promotes INS expression in vitro. a** Human Wk16.3 pancreas stained for WNT5A and PECAM1 (left panel) or VIM (right panel). DAPI marks nuclei. Scale bar = 100 μm. N = 4 independent experiments. **b** Quantification of PECAM1+WNT5A+ or VIM+WNT5A+ cells in Wk16 pancreas in percentage of total PECAM+ or VIM+ cells, respectively. Data are shown as mean ± 95% CI. N = 4 independent experiments. **c** hESC-derived EPs after ectopic *WNT5A* expression (pWNT5A), were stained for WNT5A and INS. DAPI marks nuclei. Backbone plasmid served as mock control (pCDNA). Scale bar = 50 μm. N = 4 independent experiments. **d** Quantification of INS+ cells after ectopic *WNT5A* expression (pWNT5A) in EPs. Mean ± 95% CI and p values (unpaired two-tailed *t*-tests vs. pCDNA control) are shown. N = 4 independent experiments.) **e** WNTA KO in M-E17 cells. WNT5A KO in M-E cells was confirmed by immunofluorescence for WNT5A and control staining for VIM. DAPI marks nuclei). Scale bar = 100 μm. N = 3 independent experiments. See Supplementary Fig. 8C, D for KO design. **f** M-E20 or WNT5A KO (W5A KO) M-E cells were cocultured with EPs for 4 days, and INS+ cells were quantified from IF and normalized to no coculture control (EP W/O). EP and M-E20 coculture increased INS+ cell number by 4.5-fold compared to EP W/O, while coculture with WNT5A KO M-E caused 2-fold decrease in INS+ cells for both pairs of sgRNAs used to target WNT5A locus (KO1 + 2 or KO3 + 4). The low induction of INS+ cells with WNT5A KO M-E cells was partially rescued by WNT5A recombinant protein (blue). Mean ± 95% CI and p values (Tukey's multiple comparisons test) are shown. N = 3 independent experiments. **g** Spheroids consisting of EPs and wild-type or WNT5A KO (W5A KO) M-E cells were stained for INS and VIM, showing induction of INS expression only in presence of control M-E cells, while only few cells express INS in the presence of WNT5A KO cells. Scale bar = 50 μm. N = 6 independent experiments.

expression and phosphorylation of JNK in EPs as determined by western blot (Fig. 6h). The downstream JNK effector, c-JUN, was also hyper-phosphorylated (a nine-fold increase compared to untreated control) after short-term WNT5A treatment (Fig. 6i, j). Interestingly, the activated c-JUN was mostly present in INS+ cells, suggesting the rapid induction of β-cell identity upon short-term WNT5A treatment and JNK involvement in this process.

We therefore assessed the JUN/PCP pathway as a putative downstream effector of WNT5A during in vitro β-cell differentiation. We inhibited JNK in EPs by small molecule SP600125, which resulted in diminished c-JUN phosphorylation as confirmed by western blot (Supplementary Fig. 10C). Significantly, use of the JNK inhibitor reduced the number of INS+ cells in a dose-dependent manner, indicating that JUN/PCP pathway plays an important role in β-cell induction (Fig. 6k). We also found that WNT5A addition to the EPs cultured with SP600125 cannot counteract the actions of the JNK inhibitor, suggesting WNT5A to act upstream of JNK/c-JUN (Fig. 6k). Together, these data show WNT5A activates the non-canonical WNT and c-JUN/PCP pathways during EP differentiation into β-cells.

**Long-term WNT5A treatment leads inhibits BMP signaling.** Besides JUN/PCP pathway upregulation by WNT5A, RNA-seq data analysis revealed a potential link between WNT5A signaling and BMP suppression during in vitro β-cell differentiation. The WNT5A treatment of EPs for 5 days downregulated multiple components of BMP pathway, including *BMP3, 4* and *6* as well as *GDF5* and *9*, and upregulated expression of *BMPER* and *DCN*, encoding BMP antagonists (Fig. 7a), which was further verified by qRT-PCR (Fig. 7b). Of note, according to our RNA-Seq data from M-E cells, *BMPER* and *DCN* are also enriched in β-cell fate-promoting M-E17h and M-E20 as compared to ineffective M-E9 (Supplementary Fig. 11). Therefore, we investigated whether crosstalk between the BMP and WNT5A pathways contributes to the β-cell formation. First, we performed a triple luciferase reporter assay (Fig. 7c). We postulated that WNT5A treatment causes sequential activation of AP-1 transcription factor complex (comprising e.g., c-JUN) and downregulation of Smads 1, 4, and 5 as well as ID1, 2, 3, and 4, downstream effectors of BMPs. We therefore created a multigenic construct that included two transcriptional units to monitor two biological pathways in the same cell population. The vector contains four Smad-binding elements (4xSMAD) upstream of a minimal synthetic promoter to drive the expression of the Red Firefly luciferase (RedF), followed by six copies of the AP-1 binding element (6xAP1) and a minimal promoter driving the expression of the Firefly luciferase (FLuc) (Fig. 7c). To normalize the data between biological replicates, we included the standard reporter, Renilla luciferase, driven by the

Cytomegalovirus (CMV) enhancer and promoter. Using the construct, we found that 24-h WNT5A treatment induces AP-1 activity in EPs, followed shortly by a decrease in Smad activity at 36 and 48 h (Fig. 7d).

Phosphorylation activates Smads and results in their translocation from cytoplasm to nucleus[80]. We therefore tested for Smad1/5 activation in hESC-derived β-cells using IF (Fig. 7e, f) and single-cell post-FACS imaging (Fig. 7g). We observed a correlation between the cellular localization of phosphorylated Smad1/5 (p-Smad1/5) and INS expression (Fig. 7e–g). About 80% of cells with high INS protein levels had p-Smad1/5 localized in the cytoplasm, whereas p-Smad1/5 was predominantly in the nucleus in cells with low or no INS expression (Fig. 7e–g), suggesting that BMP and Smad activity might inhibit INS expression.

To mimic an in vivo source of BMP inhibition, we treated EPs with Gremlin1 (Grem1), a known BMP antagonist[81] (Fig. 7h) expressed in M-E cells. We found that a 3-day treatment with 200 ng/ml Grem1 led to a 5-fold increase in both CHGA+ and INS+ cell numbers, while the combination of 500 ng/ml WNT5A and 200 ng/ml Grem1 increased CHGA+ by 17-fold and INS+ by 16-fold as assessed by IF (Fig. 7h, i). Moreover, β-cells derived by a 5-day EPs treatment with WNT5A alone or in combination with Grem1 had increased glucose responsiveness and insulin content after 14 days culture, as compared to untreated controls (Fig. 7j); yet the treatments did not reach effectiveness of coculture with M-E20. Interestingly, single WNT5A or Grem1 or co-treatment caused a 1.5-fold decrease in the number of GCG+ cells, which are a common byproduct of β-cell in vitro differentiation (Supplementary Fig. 12). Summarizing, WNT5A treatment led to activation of the JNK/c-JUN/AP1 followed by a decrease in BMP signaling as well as upregulation of CHGA and INS while suppressing GCG, which was further enhanced by addition of BMP signaling inhibitor, Gremlin1. Together, our results indicate that human fetal pancreatic M-E cells are a source of WNT5A and inhibitors of BMP and WNT-JNK-BMP signaling crosstalk between M-E and EPs specifically promotes β-cell formation (Fig. 7k).

## Discussion
Tremendous efforts have been implemented into improving the efficiency of pancreatic cell in vitro differentiation. However, many gaps remain in our understanding of the molecular mechanisms regulating pancreatic endocrine cell specification, including the complex role of microenvironment. Therefore, we reconstituted the human pancreatic microenvironment from various stages of development and used M-E cells to support the efficient β-cell generation from hPSCs. We found that coculturing M-E17 and M-E20 cells with hPSC-derived PPs or EPs leads to a

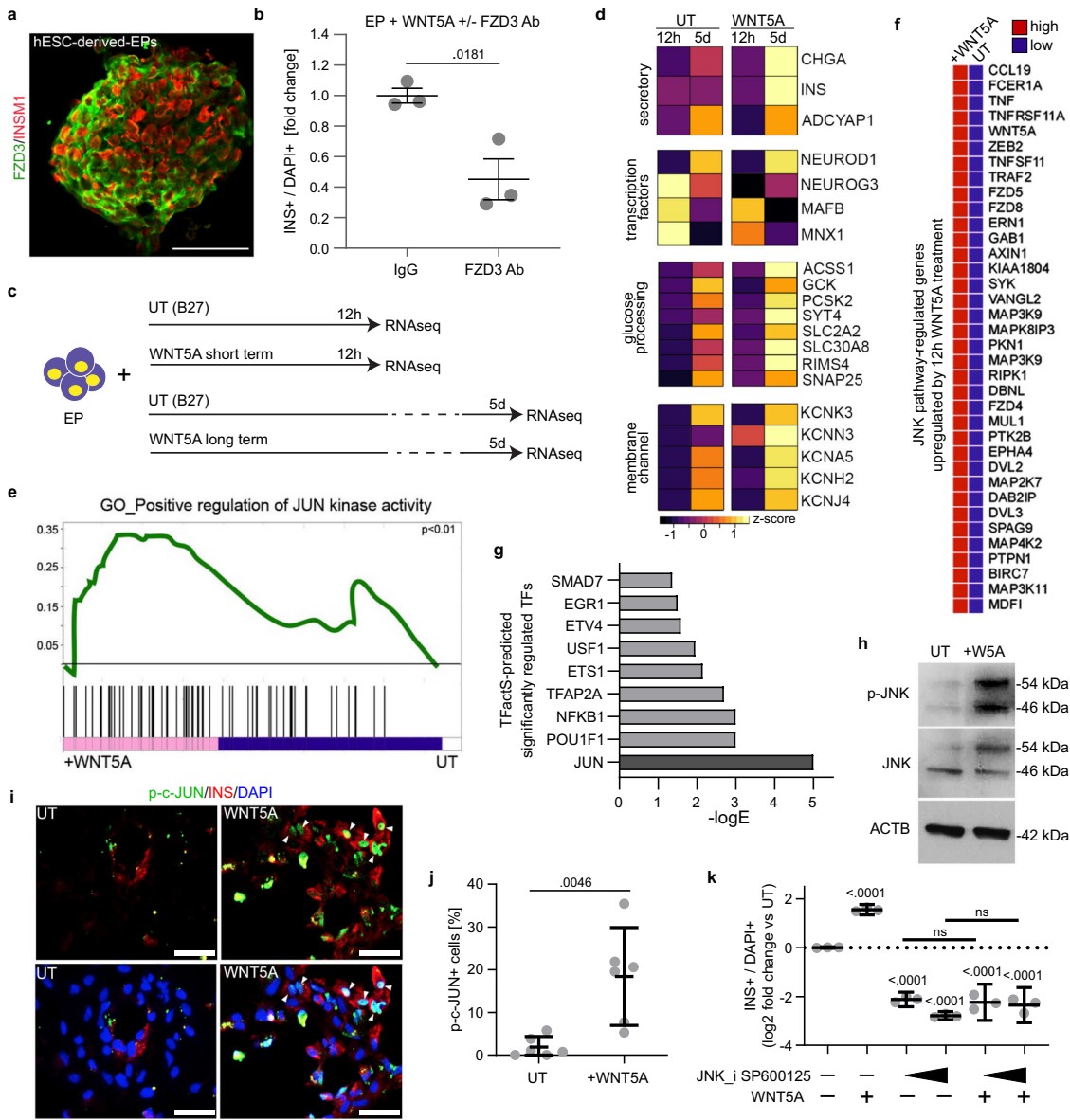

**Fig. 6 Short-term WNT5A treatment activates the JNK/c-JUN pathway in human EPs. a** hESC-derived EPs stained for FZD3 and INSM1. Scale bar = 50 μm. N = 3 independent experiments. **b** Blocking FZD3 receptor in EPs leads to ~2.5-fold decrease in INS+ cells compared to untreated control at day 3. Mean ± SEM and a p value (unpaired two-tailed Student's t test) are shown. N = 3 independent experiments. **c** Scheme to investigate by RNA-sequencing global gene expression changes in human EPs induced by short- (12 h) and long-term (5 days) WNT5A treatment. **d** WNT5A treatment shifts the transcriptional profile of hESC-derived EPs toward that of β-cells. Z-score of normalized RNA-sequencing expression values of selected genes with at least one pairwise difference of q < 0.05. **e** GSEA plot from RNA-sequencing showing short-term EP treatment with WNT5A activates JUN pathway. **f** GSEA heatmap showing JNK target upregulated genes in hESC-derived EPs after short-term WNT5A treatment. **g** Predicted significantly upregulated TFs in EPs after short-term WNT5A treatment, showing high c-JUN upregulation as analyzed by TFactS. **h** Western blot for total and phosphorylated JNK after WNT5A treatment of EPs. Beta-actin is a loading control. Multiple bands of p-JNK and JNK correspond to 54 and 46 kDa isoforms. N = 3 independent experiments. **i** Phospho-c-JUN and INS staining in EPs treated for 24 h with WNT5A or in untreated (UT) control. DAPI marks nuclei. Scale bar = 100 μm. N = 6 independent experiments. **j** Quantification of p-c-JUN+ cells shown are a percentage out of total (DAPI+) cells in untreated (UT), and WNT5A-treated EPs showing a 6-fold increase in the number of c-JUN+ cells. Mean ± 95% CI and a p value (unpaired two-tailed t-test) are shown. N = 6 independent experiments. **k** JNK inhibition by SP600125 lowers INS+ cell induction. EPs were treated for 3 days with 20 μM or 40 μM SP600125, and INS + were quantified and presented as fold change compared to vehicle (DMSO) treated control. Co-treatment of EPs with SP600125 and WNT5A (blue) does not rescue INS+ cell induction. Mean ± 95% CI and p values (Tukey's multiple comparison test vs. control) are shown. N = 5 independent experiments.

robust increase in β-cell formation and glucose responsiveness, approaching that of human islets[82]. This effect was achieved without factors that are usually added to medium at this stage in various differentiation protocols. Interestingly, M-E17 and M-E20 cells were derived from stages of the second wave of human β-cell formation, whereas the non-effective M-E9 cells were derived

after the first wave β-cells had already emerged[45]. Therefore, our findings underscore the importance of using age-matched pancreatic microenvironment for research on its role in islet cell development. For example, the pancreatic niche from later developmental stages might be a source of unknown signals for β-cell maturation or maintenance. Inversely, using PPs

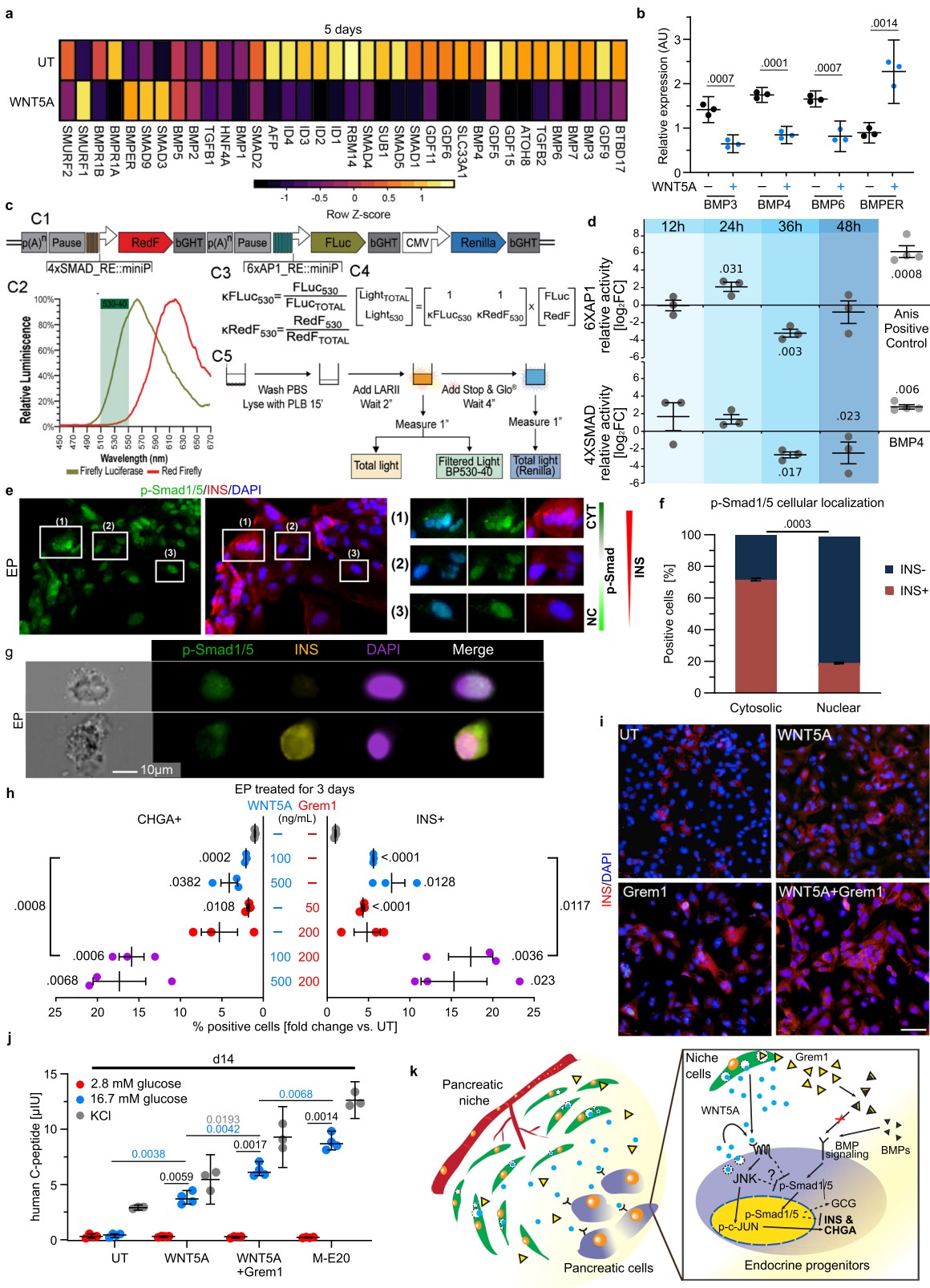

differentiated for a shorter time (e.g., 8 days instead of 9 days) in coculture with M-E9 could induce first wave-like endocrine differentiation if it is triggered by M-E factors.

Next, we found that both ECM and soluble factors secreted by M-E cells promote β-cell fate decisions. Of the tested factors, we found WNT5A, expressed in human developing pancreas' niche in vivo, has the most robust effect on β-cell induction in vitro. We

confirmed these results are independent of the hESC cell line or differentiation protocol, as well as by WNT5A overexpression, anti-WNT5A antibodies, and coculture with WNT5A-KO M-E cells combined with rescue experiments.

Mesenchyme-derived Wnt5a has already been identified as important for pancreatic islet development in zebrafish and mice but not in humans[83,84]. In these studies, overexpression of Wnt5a

**Fig. 7 Long-term WNT5A treatment inhibits BMP signaling in hESC-derived EPs. a** BMP pathway-associated gene expression after 5 days WNT5A treatment of EPs compared to untreated (UT) control. Downregulation of BMP ligands, effectors and BMP antagonist upregulation suggests decreased BMP activity. Log$_2$ Z-score of normalized RNA-sequencing expression values are shown. **b** qRT-PCR verification of selected gene expression after 5-day WNT5A treatment. Mean ± 95% CI and p values (unpaired two-tailed t-tests) are shown. N = 3 independent experiments. AU arbitrary units. **c** Dual pathway luciferase system to assess activity of Jun (AP1) and Bmp (Smad) pathways. Details in "Methods". **d** Quantification of relative AP1 and Smad activity in WNT5A-treated EPs. Anisomycin (Anis) and BMP4 positive controls for AP1 and Smad, respectively. Mean ± SEM and p values (unpaired two-tailed t-test) are shown. N = 3 independent experiments. **e** Immunofluorescent analysis of phospho-Smad1/5 cellular localization in INS+ cells. p-Smad1/5 is mostly in cytoplasm (CYT) of INS+ cells, while in nucleus (NC) in INS- cells. N = 4 independent experiments. **f** Quantification of cytoplasmic and nuclear p-Smad1/5 in INS+ and INS- cell. Mean ± SEM and a p value (unpaired two-tailed t-test) are shown. N = 3 independent experiments, ≥10 images per experiment. **g** post-FACS imaging of single EPs to detect p-Smad1/5 and INS cellular localization. The majority of cells expressing nuclear p-Smad1/5 (top) do not express INS, N = 4 independent experiments. **h** Quantification of CHGA+ and INS+ cells after 3-day EP treatment with Gremlin1 (Grem1), WNT5A or in combinations. Mean ± SEM and p values (unpaired two-tailed t-tests) are shown. N = 3 independent experiments. **i** INS immunostaining of EPs treated for 3 days with WNT5A, Grem1, and Grem1 + WNT5A. Scale bar = 50 μm. N = 6 independent experiments. **j** GSIS analysis after 5-day EP treatment with WNT5A alone or in combination with Grem1 compared to untreated EPs (UT) or M-E20 coculture. Graph represents ELISA measurements. N = 4 (glucose) or 3 (KCl) independent experiments. Mean ± 95% CI, and significant p values (Tukey's multiple comparison test) are shown. **k** Model of WNT5A role in human pancreatic niche during β-cell differentiation. Pancreatic stage-specific M-E cells secrete WNT5A and Grem1, which activate JNK/c-JUN/AP1 and inhibit BMP pathways leading to CHGA, INS upregulation and GCG downregulation in hESC-derived EPs. DAPI marks nuclei.

in Pdx1+ epithelium in mouse[83] and *Wnt5a*-KO in zebrafish and mouse[84] resulted in defective migration and clustering of β-cells into islets. However, the altered *Wnt5a* expression did not seem to affect the number of endocrine cells. Conversely, ceased expression of mesenchymal *Wnt5a*, as a result of the upstream regulator *Hox6* knock-out in mouse, yielded almost completely hindered EP-to-endocrine transition[85]. The effect was rescued in pancreatic explants exposed to recombinant Wnt5a, which is in line with our observations of pro-endocrine WNT5A function in human EPs. The discrepancy could be explained by redundancy in functions of WNT5A and WNT5B, with the latter being present in the *Wnt5a*-KO mice[84] but being downregulated in the *Hox6*-KO mice[85]. In our scratch assay and transwell assay experiments, which addressed the pro-migratory role of WNT5A, we did not observe significant effects. It is however possible that during the 3D hPSC in vitro differentiation delaminating EPs and early endocrine cells do not migrate extensively but rather form as endocrine clusters, as suggested in the recently proposed "peninsula" model of islet formation[86]. In addition, there is no mesenchymal microenvironment present during in vitro differentiation so emerging endocrine cells might not be able to migrate through it. Therefore, development of more sophisticated tools might be necessary to evaluate hPSC-derived β-cell migration and detailed insight into WNT5A role in human EPs migration or clustering in vitro is still pending.

Here we focus on the molecular mechanism of WNT5A-mediated human pancreatic endocrine development. WNT5A can act through both canonical and non-canonical Wnt signaling, and through multiple receptors and we show that WNT5A utilizes a non-canonical pathway during human EP differentiation. In postnatal mice, the non-canonical Wnt/PCP pathway triggered β-cell maturation through its effector and reporter Flattop (*Fltp*)[87]. However, in our system we did not observe altered *FLTP* expression. Yet, FLTP might be involved in late-stage β-cell maturation rather than β-cell fate acquisition, as in the case of our experiments. In the work of Bader et al.[87] WNT5A increased insulin secretion of human microislets in vitro, whereas Vethe et al.[88] applied WNT5A and B together on late stage (S7) β-cells and observed increased number of bi-hormonal INS+/GCG+ cells. In contrast, we show accelerated specification of EPs into functional, glucose-responsive (Fig. 7j) and GCG negative (Supplementary Fig. 12) β-cells. The discrepancy might result from stage-specific response to WNT5A as we have tested its influence on β-cell fate acquisition from EPs, while the others in mature or maturing β-cells. Further, WNT5A might act through different mechanisms that we uncovered in our research, while they were not shown directly in former works.

The migratory function in zebrafish was mediated by fz-2[84], yet we did not observe any effect of WNT5A on human EP migration and our experimental results indicate that specifically in the context of endocrine differentiation into β-cells, the ligand-receptor axis is WNT5A-FZD3. FZD3 was shown to function in the PCP pathway in the nervous system, where it interacts with Celsr3[89–91]. Interestingly, genetic loss of Celsr2 and 3 in mice decreases the number of INS+ cells and disrupts the JNK pathway[92], which together with our finding that WNT5A activates JNK/c-JUN/AP1 pathway in EPs suggests a plausible mechanism of WNT5A-induced β-cell commitment.

Niche-derived BMP has a complex, context-dependent role in pancreatic development[31,93,94]. While promoting progenitor proliferation and maturation, BMP suppression is required to induce endocrine specification[93,95,96]. The upstream mechanisms tightly regulating BMP signaling during pancreatic development and function remain vague. We found that long-term WNT5A treatment inhibits BMP signaling and that M-E cells-derived BMP inhibitor Grem1 facilitates β-cell specification. Interestingly, we show that WNT5A induces the INS+ cells without additional BMP inhibition. Importantly, blocking BMP too early in PDX1+ PPs, prior to NKX6-1 expression, results in precocious endocrine cell differentiation, specifically promoting α-cell fate. Indeed, the efficient α-cell differentiation protocols include early BMP inhibition[97–99]. Interestingly, WNT5A and BMP are opposedly involved in pancreatic vs. liver specification, where WNT5A promotes pancreas[100] whereas BMP promotes liver[94,101], which further suggests possible inhibition of BMP by WNT5A within the pancreatic niche. How WNT5A suppresses BMP requires further investigation, but this can be mediated by Jun/AP1 pathway as these pathways crosstalk in a complex, unclear way[102–105].

While we focused on signals that contribute to β-cell development, adult functional islets are composed of multiple endocrine and non-endocrine cell types that tightly cooperate to dynamically regulate blood glucose levels. Therefore, a successful β-cell replacement therapy approach might require manufacturing of islet-like structures composed of multiple cell types. There are no protocols for efficient derivation of other pancreatic endocrine cells than β-cells and α-cells, mainly because developmental signals driving these cells formation remain unknown. Here, we identified Endocan, a proteoglycan which can bind integrins[106] and which is involved in angiogenesis[107], as a factor specifically promoting δ-cell formation. Interestingly, EPs treatment with Endocan induced SST+ cells with efficiency exceeding the induction during the development, and to our knowledge this

is the first growth factor identified to promote SST+ cell fate. The mechanisms beyond this specific Endocan function remain to be identified.

## Methods

**Ethics statement.** To derive M-E cells, we obtained human fetal pancreas, duodenum, and spleen, from 9.1 to 20.1 weeks after fertilization, under Institutional Review Board guidelines at Baylor College of Medicine—H-3097 to M.B. No compensation was provided for the participants. Human islets were obtained under Human Islet Isolation for Research (HIIR) protocol: Pro00001097 to O.M.S. at Methodist Research Institute. All samples were deidentified and we did not obtain any personal information. No compensation was provided for the participants. The study design and conduct complied with all relevant regulations regarding the use of human study participants and was conducted in accordance with the criteria set by the Declaration of Helsinki. Animal studies (MEF derivation) were regulated and approved by the IACUC at Baylor College of Medicine, protocol AN-6325 to M.B.

**Animal studies.** ICR/CD1 6-12 week females purchased from Taconic Biosciences for MEF derivation from e12.5 embryos to derive MEFs. MEFs were derived from 18 embryos.

**Derivation of human M-E primary cells.** To derive M-E cells, we obtained human fetal pancreas, duodenum, and spleen, from 9.1 to 20.1 weeks after fertilization, under Institutional Review Board guidelines. We dissected each tissue into ~4 mm³ cubes. Samples were transferred to 6-well plates and kept at 37 °C for 10 min to allow attachment of the tissue to the plate surface. Then, DMEM:F12 medium (+10% FBS, 1× penicillin-streptomycin, 1× Glutamax (all Invitrogen)) was added. Over the subsequent 2 weeks, medium was changed every other day, and wells were monitored for outgrowth of M-E cells. Once 50% confluent, the cubes were removed, and M-E cells were dissociated using 0.25% trypsin-EDTA, and expanded for banking. For each stage, there were at least two independent cell derivations completed. In addition, previously established cells were used: human dermal fibroblasts (HDFs, ATCC), human umbilical vein endothelial cells (HUVECs, ATCC), MEFs (E12.5 ICRs, Taconic) and mouse islet endothelial cells (MS1, ATCC).

**hESC maintenance and pancreatic differentiation.** hESC lines, ISL1-EGFP (a gift from Drs Kenneth Chien and Lei Bu)[67], Hues8 (Harvard Stem Cell Institute) and H1 (Wicell, WA-01), were maintained under a feeder-free system on Geltrex (Invitrogen) in TeSR-E8 media (Stemcell Technologies) We passaged cells at 70–80% confluence using TrypLE (Invitrogen). After dissociation, cells were seeded in TeSR-E8 media + 10 µM Y-27632 for 24 h. Pancreatic differentiation was started when cells were 90% confluent while still maintaining their colony shape with defined borders.

The following media and growth factors/small molecules (see also Supplementary Table 3 for concentrations) were used for pancreatic differentiation unless otherwise noted: day 1: MCDB-131 media with 0.1% BSA + 10 mM glucose + ActivinA and CHIR99021. Days 2–3: MCDB-131 media + 0.1% BSA + 10 mM glucose + ActivinA. Days 4–5: MCDB-131 + 0.1% BSA + 10 mM glucose + Vitamin C (VitC) + KGF. Days 6–9: MCDB-131 + 2% BSA + 5.5 mM glucose + VitC + Insulin-Transferrin-Selenium (ITS) (Invitrogen) + ActivinA + KGF + RA + SANT-1 + Noggin. Days 10–12 (PP stage): MCDB-131 + 2% BSA + 5.5 mM glucose + VitC + ITS + SANT-1 + Noggin + PdBU. Days 13–15 (EP stage): MCDB-131 + 2% BSA + 5.5 mM glucose + VitC + ITS + Noggin + AlK5i. To test the role of WNT5A in EP development, we used several protocols to derive PPs based on Pagliuca et al.[9], Nostro et al.[108], McGaugh et al.[109] and Russ et al.[66] as outlined in Supplementary Fig. 7E. Differentiation protocol details and efficiencies at intermediate stages are shown in Supplementary Fig. 7E. We applied these protocols to ISL1-eGFP, H1 and Hues8 hESC lines.

In experiments testing selected growth factors effects on endocrine fate induction, EPs were dissociated and seeded as 25,000 cells/well on Geltrex-coated 96-well plates in DMEM media (Invitrogen) with B27 Supplement (Thermo Fisher Scientific), later called B27 media, supplemented with 10 µM Y-27632 for 24 h. Growth factors (Supplementary Table 3) were added to B27 media, at two concentrations, for 3 days, with everyday medium change. After 3 days, cells were PBS washed, fixed with 4% PFA/PBS for 20 min at room temperature (RT), washed twice with PBS and stained.

**Coculture of hESC-pancreatic progenitors and M-E cells or M-E-derived ECM or CM.** Three settings were used for M-E cells and hESC-derived PP interactions. The first was direct cell-cell interaction, the second was a culture of PPs or EPs in CM collected from M-E cells, and the third was a culture of progenitors on the M-E-derived ECM. For the first assay, M-E cells were plated on 6-well plates 24 h in advance. Mitotic inactivation was performed for 2 h with 10 µg/ml mitomycin C (Sigma-Aldrich) followed by three washes with PBS. In the meantime, hESCs were differentiated to either the PP or EP stage and plated on M-E cells at a density of 60,000 cells per cm². As controls, plates coated with Geltrex (W/O condition) or

inactivated HDFs, MEFs, HUVECs or MEFs + HUVECs were used. CM was prepared from 60% confluent M-E cells cultured in the same media as for hESC differentiation, and media were collected every day for 6 days. The collected CM was later used to differentiate PPs or EPs. as control CM was heat-inactivated at 85 °C for 10 min. For the third assay, M-E ECM plates were prepared as follows: confluent M-E cells were cultured on 6-well plates for 6 days, after which cells were removed by short non-enzymatic EDTA treatment, leaving the ECM behind. PPs or EPs were plated on these ECM-plates and differentiated into β-cells.

For coculture of M-Es with EPs in 3D format (Fig. 5g) hESCs were cultured and differentiated into EPs using the Pagliuca protocol. EPs were dissociated and combined with mitomycin C-treated M-E cells in ratio 4:1 and plated into 24-well low-attachment plates at cell density 2.5 × 10⁶ cells/ml in DMEM media supplemented with B-27 and 10 µM Y-27632. Cells were incubated with rocking (70 rpm) in 37 °C for 4 days. After 4 days, spheroids were fixed and processed for whole mount staining as described in the immunofluorescent analysis section.

**Human islet isolation.** Human pancreata were obtained with informed consent for transplant or research use from relatives of heart-beating, cadaveric, multi-organ donors through the efforts of The National Disease Research Interchange, Tennessee Donor Services, the Mid-South Transplant Foundation, and the United Network for Organ Sharing. Human islets were isolated from six non-diabetic donors. Donor demographics were collected at the time of acceptance and included age in years, gender, body mass index and cause of dead, which are provided in Supplementary Table 5. Pancreas was perfused using the University of Wisconsin (UW) solution. Human islets were isolated from cadaver donors using an adaptation of the automated method described by Ricordi et al.[110]. Liberase (Boehringer Mannheim, Indianapolis, IN) was the enzyme used in all the isolations in this study. Liberase was dissolved in cold (4 °C) Hank's balanced salt solution (HBSS) (Mediatech, Inc., Herndon, VA) that was supplemented with 0.2 mg/ml DNase (Sigma Chemical Co., St. Louis, MO), 1% penicillin-streptomycin (Sigma Chemical Co.), 20 mg/dl calcium chloride (J.T. Baker, Inc., Phillipsburg, NJ), and HEPES (Sigma Chemical Co.). Once dissolved, the pH was adjusted to between 7.7 and 7.9. The enzyme preparation was then sterile filtered, warmed to 37 °C, and used for the intraductal distension of the pancreas. The distended pancreas was cut into several pieces, placed in the Ricordi's chamber, and the Heating circuit started. Pancreatic digestion was performed at 37 °C until more than 90% free islets were observed in the sample. Digested tissue was collected into cold HBSS supplemented with 20% human serum and 1% penicillin-streptomycin solution and centrifuged at 400 × g at 4 °C for 5 min. Tissue pellets were pooled into a cold UW solution and held at 4 °C for 1 h with periodic mixing. Islet purification was performed on a COBE 2991 Cell Processor (COBE BCT, Lakewood, CO) using OptiPrep (Nycomed Pharma AS, Oslo, Norway) as a step-gradient based on a modification of the procedure of Robertson et al.[111]. Islet culture: Aliquots from human islet isolations were cultured in SFM containing 1% ITS, 1% L-glutamine (Life Technologies, Gaithersburg, MD), 1% antibiotic-antimycotic solution (Sigma Chemical Co.), and 16.8 µM zinc sulfate. ITS (1%; Collaborative Biomedical Products, Bedford, MA[112]). Islets were seeded in 804G-coated 96-well plates and incubated in CMRL1066 media (Mediatech Inc.) supplemented with 10% human serum overnight. After 3 days, we performed GSIS.

**GSIS.** Cells were washed with Krebs buffer (128 mM NaCl, 5 mM KCl, 2.7 mM CaCl₂, 1.2 mM MgCl₂, 1 mM Na₂HPO₄, 1.2 mM KH₂PO₄, 5 mM NaHCO₃, 10 mM HEPES, 0.1% BSA) and then pre-incubated in 2.8 mM D- (+)-glucose Krebs buffer for 2 h. Cells were then incubated in fresh-low glucose Krebs, followed by 16.7 mM and then in Krebs buffer with 2.8 mM glucose and 30 mM KCl for 30 min at each condition. After each incubation, we collected supernatants. Between incubations, cells were washed two times with Krebs buffer. This procedure was repeated at least three times for different time points and coculture combinations. In the end, cells were dispersed into single cells using TrypLE Express and quantified by Countess (Invitrogen). C-peptide was measured using the Human Ultrasensitive C-peptide ELISA (Mercodia). The C-peptide amount was normalized to the cell number (µIU/3000 cells).

**mRNA sequencing.** Total RNA was extracted from two biological replicates of each sample using Trizol method with DNase treatment, and RNA quality was assessed using 2100 Bioanalyzer (Agilent Genomics). Samples with RIN ≥ 9 proceeded to library preparation using TruSeq stranded mRNA Library Prep Kit LT (Illumina) according to the manufacturer's protocol. Library concentration was determined by qRT-PCR (KAPA Library Quantification Kit) to pool equal amounts of libraries with different adapter indexes. Sequencing was performed using NextSeq500 (Illumina). Data were analyzed as described in Extended Experimental Procedures.

**Dual-pathway luciferase vector and multicolor luciferase assay.** We built the dual-pathway luciferase reporter vector using the GoldenBraid2.0 Assembly Platform[113,114]. Briefly, transcriptional units comprising the promoter elements, the CDS of the corresponding luciferase and the bovine growth hormone terminator were first assembled, and then these were later combined to build the multigenic vector used in this assay. All elements are included in the same DNA

string ensuring the simultaneous transfection of all the reporters in equal ratios in the cells. Four copies of the Smad binding element (4xSmad_RE) were cloned upstream of a synthetic minimal TATA-box promoter with low basal activity (miniP) to drive the expression of the *Red Firefly luciferase* (RedF). Six copies of the AP-1 binding element (6xAP1) were assembled upstream of the miniP to drive the expression of the *Firefly luciferase* (FLuc). The CMV enhancer and promoter drove the expression of the standard luciferase *Renilla*. Each transcriptional unit included the bovine growth hormone terminator (bGHT) and a synthetic polyA-p(A)n- and a transcriptional pause signal –Pause- were added upstream of the DNA response elements to prevent interference derived from the transcription of the upstream luciferase. For the multicolor luciferase assay, H1-derived EPs were first dissociated into 96-well plates as earlier described and incubated overnight to allow cell attachment. Transient transfections were then performed using 0.75 µl Lipofectamine 2000 with 150 ng of dual-pathway luciferase vector for each well of the 96-well plate and incubated for 24 h. Positive controls (CMV:FLuc:bGHT, CMV:RedF:bGHT and CMV:Renilla:bGHT) were transfected in separate wells to adjust the transmission constants for each luciferase. Transfected EPs were further treated with 500 ng/ml WNT5A, 200 ng/ml BMP4, 1 ng/µl Anisomycin dissolved in basal media, as previously described. At the determined harvesting point, culture media was removed, and cells were washed with PBS. In total, 35 µl of passive lysis buffer was added to the wells. Culture plates were incubated at RT for 15 min on a rocking platform and stored at −80 °C for further assay until all data points were collected. After thawing the lysates, they were transferred to a 384-well plate, and the luciferase assay was performed in a CLARIOstar illuminometer. In total, 10 µl of LARII reagent was added with the built-in injectors, and after 2 s, the total light and the BP filtered light emitted by the FLuc and RedF mixture were measured for 1 s. Finally, 15 µl of Stop & Glo® reagent was injected, and after 4 s the emitted light by Renilla luciferase was measured. The activity corresponding to FLuc and RedF that were simultaneously measured after the LARII reagent was added were calculated according to the method proposed by Nakayima et al. [115] that is adjusted to this particular assay and explained by a scheme in Fig. 7c, and is described here: (C1) The multigenic construct used in the dual-pathway luciferase assay, with Smad binding elements (4xSmad_RE) regulating expression of the Red Firefly luciferase (RedF), AP-1 binding elements (6xAP1) driving expression of the Firefly luciferase (FLuc), and the Renilla luciferase under the CMV promoter (see "Methods" for the extended description). (C2) Spectra of Firefly (FLuc) and Red Firefly (RedF). The 530-40 bandpass filter (BP) used for the luciferase measurement is indicated over the spectra. (C3) The transmission constants for each luciferase (κFLuc530 and κRedF530) were calculated by dividing the transmitted light (FLuc530 and RedF530) by the total light emitted by each luciferase (FLucTOTAL and RedFTOTAL). (C4) An equation for simultaneous calculation of luciferase activity in the Red Firefly and Firefly Luciferase mixture is shown. LightTOTAL is the total relative light units (RLU) measured in the absence of the optical filter, FLuc530 are the RLU of FLuc that pass through the BP, RedF530 are the RLU that pass through the 530-540BP and FLuc and RLuc are the Firefly luciferase and the Red Firefly luciferase contribution to the mix, respectively. (C5) Overview of luciferase assay; three luciferase measurements are performed, two at 2 s after LARII reagent injection and the third one at 4 s after Stop & Glo reagent injection.

**Immunofluorescent analysis**. Cells were incubated with 5% donkey serum (Jackson ImmunoResearch Laboratories) in PBST (PBS + 0.1% Triton-X) for 30 min to avoid nonspecific antibody binding and permeabilize the cells. Primary antibodies, diluted in 5% donkey serum/PBST, were then incubated at 4 °C overnight with shaking. After primary antibody incubation, cells were washed three times with PBST and donkey secondary antibodies conjugated with Alexa Fluor 488 (i.e., Anti-Goat, Anti-Rabbit, Anti-Chicken, Anti-Mouse) or TRITC (i.e., Anti-Goat, Anti-Rabbit, Anti-Chicken, Anti-Mouse, Anti-Rat) (Jackson ImmunoResearch Laboratories) diluted at 1:400 in 5% donkey serum/PBST were added to the cells for 30 min at RT. Then, cells were thrice washed with PBST and nuclei were stained with DAPI (Roche Diagnostics). Antibody sources, catalog numbers, and dilutions are listed in Supplementary Table 2. For imaging, we used Leica DMI6000 or confocal Leica TCS SPE. Images were initially processed by LAS X software and then further analyzed and quantified using ImageJ software (NIH, W Rasband, http://rsb.info.nih.gov/ij) using cell counter plug-in. Typically, at least five randomly selected images were counted per condition and per replicate.

**Intracellular flow cytometry**. Cells were dissociated, washed with PBS, filtered through a 40 µm cell strainer and fixed with 4% PFA for 30 min at 4 °C. In total, 1% BSA, 0.1% saponin in PBS was used to dilute antibodies and to permeabilize cells. Samples were incubated with primary antibodies overnight at 4 °C rotor. After primary antibody incubation, samples were washed with 1% BSA, 0.1% saponin in PBS once and spun down at 1200 × g for 5 min. Donkey secondary antibodies conjugated with Alexa Fluor 488 (i.e., Anti-Goat, Anti-Rabbit, Anti-Chicken, Anti-Sheep) or Alexa Fluor 647 (i.e., Anti-Goat, Anti-Rabbit) (Jackson ImmunoResearch Laboratories) diluted to 1:400 were incubated overnight at 4 °C. Antibody sources, catalog numbers, and dilutions are listed in Supplementary Table 2. Then cells were centrifuged at 1200 × g for 5 min and washed 1% BSA, 0.1% saponin in PBS and filtered through a 40 µm cell strainer before flow cytometry. FACS analysis was performed using LSRII (BD Biosciences) and Diva software package. For all the

samples, at least 5000 events were captured, and FlowJo was used for gating and analysis. An example of gating strategy for cell populations and positive fluorescent staining are shown in Fig. S7c, d. Similar approach was used throughout the study.

**Imaging flow cytometry to analyze the pSmad1/5 localization**. To determine the cellular localization of pSmad1/5 and INS, one million of EPs was dissociated and filtered to single cell suspension as previously described and then fixed with 4% PFA with 0.1% Saponin for 30 min at 4 °C. After fixation, cells were centrifuged at 3000 × g for 3 min and washed with 0.1% Saponin, 1% BSA in PBS followed by incubation with primary and then secondary antibody diluted with 0.1% Saponin, 1% BSA/PBS. ImageStreamX MarkII (Millipore) was used to capture high-resolution single cell images to detect DAPI, INS, and pSMAD1/5 cellular localization. In total, 10,000 events were acquired, and compensation was adjusted to minimize spectral overlap between the fluorophores used in the experiment, which are DAPI, Alexa-488, and TRITC. Antibody sources, catalog numbers, and dilutions are listed in Supplementary Table 2. Data points were analyzed by IDEAS software (Millipore).

**qRT-PCR-based gene expression analysis**. Total RNA was isolated using TRIzol (Thermo Fisher Scientific) according to the manufacturer's protocol. DNAase (Qiagen) treatment was performed to remove genomic DNA. cDNA was synthesized using iScript (Biorad) by using 1 µg of RNA. For qRT-PCR, KAPA SYBR FAST (Kapa Biosystems) was used and the reaction was run in Connect CFX light cycler (Biorad). Primers were designed using qPrimerDepot in such a way that the PCR product spans across exons junctions. Primer sequences are listed in Supplementary Table 1. Primer specificity was checked using melting curve analysis in CFX manager software v3.1 (Applied Biosystems) and PCR product electrophoresis. Threshold data were analyzed by CFX manager software v3.1 using Comparative Ct relative quantitation method with TBP as an internal control.

**RNA-sequencing data analysis**. After preliminary analyses, showing a significant Pearson correlation coefficient for gene expression, two biological replicates were combined and analyzed as single samples. Sequencing reads were first aligned using TopHat, and the gene differential expression was assembled and analyzed using Cufflinks. Significantly up- and downregulated genes were determined by comparing Fragments Per Kilobase of transcript per Million mapped reads between M-E and control lines, or between WNT5A-treated and untreated control, with p <0.05. Genes upregulated >2-fold and downregulated <0.5-fold were used to generate Venn diagram from BioVenn (http://www.biovenn.nl/) and the gene function categorization was refer from Hrvatin et al. [116]. To predict which transcription factors are responsible for the identified gene expression changes, TFactS analysis (http://www.tfacts.org/TFactS-new/TFactS-v2/index1.html) was performed by inputting up- and downregulated gene lists. Significantly regulated transcription factors were determined with p, e, and q < 0.05, as the default setting of the software. For GSEA (http://software.broadinstitute.org/gsea/index.jsp), all input files were generated through GenePattern and the analysis was performed based on the instruction from Broad institute GSEA user guide with the following parameter: phenotype labels as 5dUT vs. 5dWNT5A, 1000 genes set of permutations, weighted enrichment statistic, gene sets between 15 to 500, with log2 ratio of class as metric for ranking genes. Significantly regulated pathways have p < 0.01. Data GEO submission number: GSE90785.

**WNT5A ectopic expression and inhibition**. pCDNA-WNT5A plasmid was obtained from Dr. Marian Waterman (Addgene#35911). We used nucleofection for DNA delivery. One million EPs or HDFs were dissociated with TrypLE and resuspended in 20 µl P3 solution with supplement and 1 or 2 µg of DNA. Cells were nucleofected using Amaxa-4D nucleofector (Lonza) CM113 program. After nucleofection, we added pre-warmed medium to the cells and incubated them at 37 °C for 5 min before plating. The transfection efficiencies were evaluated using the pmaxGFP (Lonza): the efficiency was 17.5% and 15.7% for EPs and HDFs, respectively. To block WNT5A autocrine signal from EPs, 1 µg of WNT5A antibody was added to every 25,000 of EPs for 3 days before fixation and further immunofluorescence analysis.

**Generation of WNT5A KO in M-E cells**. Methods and design of WNT5A KO were described in Yang et al.[72]. Three days after the nucleofection, M-E17 and M-E20 cells were selected with 50 µg/ml G418 (Sigma-Aldrich), and the dosage was increased to 100 µg/ml after 2 days. After 8 days of selection, cells were cultured in DMEM + 10% FBS + Glutamax + β-mercaptoethanol (Invitrogen) to evaluate efficiency of KO.

**WNT5A treatment in spheroids**. In experiments shown in Supplementary Fig. 9A, B, SC-β-cells were generated using the protocol as previously described (Pagliuca et al.[9]). WNT5A were introduced into the differentiation from EP stage (EN in the original paper) together with T3, ALK5i in CMRL media for the first 2 days and then changed into T3, ALK5i in CMRL media from the 3 day. Samples were collected at the 4th day and 12th day counting from EP stage, cells were fixed with 4% PFA, and thrice washed with PBS for 10 min. For whole-mount staining,

samples were first blocked with 5% donkey serum in PBST overnight and incubated with antibodies overnight as described above.

**TOPFLASH reporter assay**. For TOPFLASH reporter assay, 80,000 EPs were transfected with 0.5 μg of TOPFLASH (Addgene #12456) or FOPFLASH (Addgene #12457) generated by Dr. Randall Moon, together with 0.25 μg pRLTK using Lipofectamine 2000 (Invitrogen) for 48 h. Cells were then treated with CHIR99021 (as a positive control), DMSO (mock control) or WNT5A for 3 days before collecting the samples. Luciferase assay was performed using Dual luciferase assay system (Promega) in which Luciferase and Renilla signals were measured by TD20/20 Luminometer (Turner Designs).

**Protein extraction and western blotting for phosphorylated JNK and total JNK**. ISL1-EGFP hESCs were differentiated into EPs and first balanced with DMEM medium with 1% BSA and NEAA) for 6 h and then t 500 ng/ml WNT5A was added. After 12 h, one milion of EPs were pelleted, PBS washed and resuspended in 250 μl of lysis buffer (10 mM HEPES pH7.5, 10 mM $MgCl_2$, 5 mM KCl, 0.1 mM EDTA pH8, 0.1% Triton X-100, 0.2 mM PMSF, 1 mM DTT, 1 tab of Complete Protease Inhibitor Cocktail (Roche)). Cell lysates were centrifuged at $12,000 \times g$ at 4 °C for 15 min, and their supernatants were collected. BCA assays were performed using Pierce BCA protein assay kit (Thermo) to determine protein concentration. For western blot, 30 μg of protein were denatured with 4x Laemmli buffer (40% glycerol, 8% SDS, 240 mM Tris-HCl pH 6.8, 5% β-mercaptoethanol, 12.5 mM EDTA, 0.04% bromophenol blue) at 95 °C for 3 min and resolved in 8% SDS-PAGE. PVDF membranes (BioRad) were used for transfer and membranes were blocked with 5% BSA in Tris-buffered saline with 1% Tween20 (TBST) for 1 h before applying primary antibody. Membranes were washed thrice for 10 min, and anti-rabbit or anti-mouse IgG-HRP (GE Life Science) at 1:1000 were added for 3 h at RT, followed by 3 washes. Antibody sources, catalog numbers, and dilutions are listed in Supplementary Table 2. HyGLO™ Quick Spray Chemiluminescent HRP Antibody Detection Reagent (Denville Scientific Inc.) was used to detect an antigen, and membranes were developed using CL- XPosure Film (Thermo Scientific). Membrane stripping was performed using a mild stripping buffer (200 mM glycine, 0.1% SDS, 1% Tween20, pH 2.2) according to the Abcam's instructions.

**EdU cell proliferation assay**. Cell proliferation was assessed using EdU-647 incorporation assay (EMD Millipore). In total, 10 μM EdU was added to untreated or WNT5A-treated EPs. Cells were fixed after 1 day of EdU incubation, and the permeabilization and detection were performed following the manufacturer's protocol. Cells were co-stained with DAPI and imaged using Leica DMI6000 microscope as previously described.

**Statistical analysis**. In all cell culture experiments at least three independent experiments were performed and for each experiment $N$ value is provided in the corresponding figure legend. Statistical significance was calculated using two-tailed Student's $t$ test or by post hoc multiple comparison tests (Dunnet's or Tukey's) after one-way ANOVA analysis, as indicated in respective figure legends. Data are presented on graphs as mean ± 95% confidence interval or SEM, as indicated in figure legends, and exact $p$ values are provided on graphs.

**Reporting summary**. Further information on research design is available in the Nature Research Reporting Summary linked to this article.

## Data availability

The data generated in this study are provided in the Source Data file. Raw RNA-Seq data generated during the study have been deposited in the GEO Repository under accession number GSE90785. Raw image files are available from the corresponding author upon a reasonable request. The authors declare that the derived M-E cells are available for the research community upon a request, from the corresponding author. Source data are provided with this paper.

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

## Acknowledgements

We thank Catherine Gillespie for critical comments on the manuscript. We thank Drs. Hoang Nguyen, Xiang Zhang, Stephanie Pangas, Kenneth Chien, Lei Bu and Joanne Richards for sharing reagents. We thank Anna Jedrzejak and all Borowiak lab members for helpful comments and discussion. This project was supported in part by the Genomic and RNA Profiling Core at Baylor College of Medicine with funding from the NIH NCI grant (P30CA125123) and the expert assistance of Dr Lisa D. White. This project was supported by the Cytometry and Cell Sorting Core at Baylor College of Medicine with funding from the NIH (P30 AI036211, P30 CA125123, and S10 RR024574) and the expert assistance of Joel M. Sederstrom. This work was supported by funding from Brown Foundation and Vivian L. Smith Foundation to O.M.S. This work was supported by the Baylor College of Medicine start-up, McNair Medical Institute, and the National Health Institute (P30-DK079638, 5 R01 AI120989, and 5T32HL092332-13), TEAM program from the Foundation for Polish Science and EU (POIR.04.00-00-20C5/16-00), the National Science Centre (OPUS UMO-2O19/33/B/NZ3/01226 and UMO-2020/39/B/NZ3/01408) to M.B. Research in the lab of KV was supported by start-up funds kindly provided by Baylor College of Medicine, the Albert and Margaret Alkek Foundation, and the McNair Medical Institute, as well as contributions from grants from the March of Dimes Foundation (#1-FY14-315), the Foundation For Angelman Syndrome Therapeutics (FT2016-002), the Cancer Prevention and Research Institute of Texas (R1313), and the National Institutes of Health (1R21OD022981, R01GM109938, and R01GM138781).

## Author contributions

Conceptualization: D.Y., J.C., and M.B.; Methodology: D.Y., J.C., A.S.P., K.J.T.V., and M.B.; Provision of human islets: O.M.S. Investigation: D.Y., J.C., M.A.S., K.W., and A.S.P.; Writing—original draft: M.B. and J.C.; Writing—review and editing: D.Y., J.C., M.A.S., A.S.P., W.J.S and M.B; Funding acquisition: M.B. and K.J.T.V.

## Competing interests

M.B., J.C., and D.Y. hold a US patent on pancreatic niche-derived factors for human endocrine development. The remaining authors declare no competing interests.
