## [Peer Review File · Nature Communications]

Human pancreatic microenvironment promotes β -cell differentiation via non-canonical WNT5A/JNK and BMP signalingReviewers' Comments:

Reviewer #1:

Remarks to the Author:

In the paper "Human pancreatic niche promotes β cell differentiation via WNT5A/JNK/AP1 and BMP signalling" Yang et al describe the function of Wnt5a in pancreatic island development and in particular in Insulin positive β -cells. In a screening attempt, the authors identify Wnt5a as a crucial factor guiding pancreatic progenitor cells to β cells. The authors demonstrate that Wnt5a is important for the pancreatic cell differentiation namely from pancreatic progenitors to endocrine progenitors to β cells. Furthermore, the authors suggest that Wnt5a activates the JNK pathway. Finally the authors suggest that Wnt5a/JNK signalling inhibits the BMP pathway in a later stage of development - especially during β cell maturation. In general the author use a wide variety of techniques to support their conclusion. However, I have some difficulties with this work: First, I feel that the function of Wnt5a signalling in pancreatic development is not a novel finding. A detailed study from Kim et al., 2005 analysed the function of Wnt5a in pancreas development already in zebrafish, Xenopus and mouse. At the end of the work, Kim et al came to the conclusion that Wnt5a is important for β cell migration and therefore for niche formation. For example Kim et al finds no reduction of insulin positive β cells in the zebrafish knock-down and in the Wnt5a ko mouse.

In the work presented in this manuscript, the authors believe that Wnt5a is important for the induction of the differentiation programme of the pancreatic niche in particular the β cells, which is fundamentally different to the conclusion of Kim et al.

Unfortunately, the authors do not discuss this discrepancy. Therefore, it is unclear if the different results can be explained by a human specific function of Wnt5a in comparison to other vertebrates. A further explanation of the difference would be based on the experimental system: Kim et al used an in vivo approach (overexpression, knock-down and knock-out in vertebrate model organisms) whereas Yang et al uses an in vitro cell culture approach with human stem cells. The author should address this discrepancy. A possibility to address it would be to add convincing human organoid studies - the xenograft model is insufficient to address this problem.

Detailed comments:

Fig2D M-E supernatant and ECM enhance β cell differentiation. The authors need to provide additional data for this claim as the experimental system does not allow this rigorous conclusion.

Fig. 2F Change orientation of the graphs

Fig 3CDE: Explain colours used in the presentation

Fig. 3 The gene expression array data should be backed up by a deep sequencing analysis.

Fig. S4 B-C The pictures do not show an increase of INS β cells after Wnt treatment. Therefore, it is difficult to believe the statistics if these are the best pictures the authors can provide.

Fig5 A The co-localisation of Wnt5a and Pecam or Vimentin is not convincing. The authors have to provide better data.

Fig 5B Furthermore a quantification is required illustrating the cells expressing both markers and cells expressing only one marker.

Fig 5E I see a 50% reduction after Wnt5a ko however the author write "a 3.6fold fewer cells".

Figure 5E The authors have to provide a rescue experiment by adding Wnt5a to the medium or by using Wnt5a transfected Wnt5a ko cells as control.

Fig 5F is not sufficient as a control experiment as it is technically different.

Fig S5D Is Wnt5a expressed in the same cells as Fzd3. How do the authors explain the results from Kim et al 2005 suggesting Fzd2 is receptor of Wnt5a in other vertebrate models?

Figure 5G-H shows no dose dependency. The authors should provide alternative data.

Figure S6A The author should provide an alternative experiment to support their claim that there is no effect on proliferation but on differentiation.

The cell migration experiments (Boyden, scratch) are not clear to me and needs a more convincing demonstration. Furthermore, the author should provide a positive controls in these experiments. Furthermore, it looks to me that the cells start to form cluster after Wnt5A treatment e.g. in Fig. 4F, 5G which do point towards a Wnt5A dependent migration effect.

Line 388 : Genes MAFB, NGN3, What is the reason behind the selection of these markers and an interpretation of this result is missing.

Fig 6E,F Could the authors add Wnt5A to overcome the effect of the JNK inhibitor? Can the effect of Wnt5a dependent stimulation of INS positive cells reverted by blocking JNK signalling? These combinatorial experiments are crucial to demonstrate that specifically Wnt5a activates the generic JNK pathway.

Fig 6S A Is the increase of pH3 cells from 1% to 2% or is the dataset normalized and this represents a relative increase from 100%? The picture in A suggests rather the second option.

Fig S6 C Scratch assay: Very difficult to see - quality of the picture needs improvement. How do the authors explain that Wnt5a treatment (highest concentration) has an opposite effect regarding cell migration? The author should provide a positive control.

Fig S6 D Boyden chamber: The author should explain why 500ng/ml of Wnt5a show a lower activity compared to Wk17.5 conditional medium. How do the authors explain the difference? The author should provide better controls.

Fig S7 It is unclear to me when the xenografts have been treated with Wnt5a. I suspect that the cells have been treated with Wnt5a prior implantation. Therefore the starting material would be different. Unfortunately it would be an in vitro expansion of the β cells and an in vivo confirmation of the maintenance of the cell fate. The title and conclusion are misleading.

General comments: Although, the small schematics in Fig. 2B,C,F are helpful, the order and the presentation of the data in this manuscript needs to be revised. In general, figures need a better description, sometimes higher-magnification of particular areas would be advantageous and some importantly - some sets of experiments are hidden in the supplementary data. The readability of the text needs to be further improved.

Reviewer #2:

Remarks to the Author:

There is something positive happening with the main claims of the paper, in that (I think) the "in vivo" study of the secreted intercellular signaling proteins coming from the human pancreatic niche (non-endodermal cells) can yield candidate proteins that increase the number of well-differentiated beta cells (insulin-producing) in in vitro protocols for the differentiation of such from the pluripotent ES or iPS lines. A problem with the m/s is that the level of care in preparing it for submission seems below par. I have annotated the m/s with a large number of notes all over the text and the figures that are designed to help. There are some real problems with some of the data, and complete inconsistencies

between the main text and the figures/figure legends. Part of the problem could be the number and density of information presented - it is a huge paper after all. I do feel that a significant amount of the data can be either removed as unnecessary - OR, the reason for its inclusion must be added. Again, some of my notes indicate these areas. The main theme and significance of the findings remain non-explicit still, in this version.

Stylistically, there are some areas where the language and grammar leave a great deal to be desired. I tried to catch some/most of these, but there are likely others. Sometimes they are quite annoying to have to read through.

My recommendation is that the authors be given a chance to look over this set of comments from me (I have not transposed them all into this box, but left them on the annotated m/s), and then prepare a much better m/s that should in some ways only then be regarded as the "first" submission. I think there are some interesting findings that are suitable for Nat Comms., but it needs some serious reworking before being able to work through it in a more full-fledged review.

Reviewer #3:

Remarks to the Author:

Summary

This manuscript describes the use of human fetal pancreatic explants to generate primary mesenchymal and endothelial cultures, which were then used to promote differentiation of human embryonic stem cell-derived pancreatic and endocrine progenitors. The mesenchymal cells in the cultures were validated with vimentin staining and endothelial cells by PECAM1 and CFIII. The authors noted that these are primary cultures that change over time, thus all experiments were done with cells within the first 6 passages. M-E cultures were used to promote the differentiation of pancreatic progenitors (PP) into insulin + cells, and were shown to be more effective than control fibroblasts. Interestingly M-E cultures from 9 week were not very effective in promoting ins+ cell production whereas 17 and 20 week M-E cells produced factors (both ECM and soluble) that enhanced both insulin expression and GSIS. The authors performed RNAseq on 17 and 20 week M-E cultures to identify candidate factors that may promote PP and EP maturation, and several of these factors were tested for their ability to enhance differentiation, in particular Wnt5a. Wnt5a is a non-canonical wnt/PCP ligand was able to substantially enhance production of ins+ cells, consistent with its role in mouse beta cell development/maturation. Wnt5a was genetically ablated from 17 week M-E cultures and was shown to be necessary for full ins-inducing activity. Wnt5a-treated cultures had reduced expression of BMP ligands and increased BMP antagonists, suggesting that repression of BMP signaling may be important for ins+ cell production. Indeed inhibition of BMP with Gremlin promoted ins+ cell numbers.

The main findings, namely the isolation and characterization of human fetal M-E cultures and their use to promote the differentiation of PP and EPs into ins+ cells, the identification and functional studies of Wnt5a in promoting differentiation of ins+ cells, are all interesting, important and very well supported by the data. There are quite a few examples where the language used and the conclusions reached are not supported by the data. There are also several examples where additional experiments would strengthen the conclusions and manuscript in general. These are all discussed in detail below.

Comments

While there is tremendous excitement over the premise of identifying embryonic populations of "niche" cells and studying how they change over time, this manuscript in its essence does not do this. The mere act of transferring cells to plastic and growing them as monolayers for months with multiple passages would dramatically change cell composition through selection. The minced tissues were

grown in DMEM:F12 with 10%FBS, which will allow for robust expansion of fibroblast-type cells, but may select against many other mesenchymal populations. The end result is that the heterogeneous population of cells at the start is likely to have a more homogeneous population of cells after months of passaging. Analysis of Vimentin and PECAM (and a handful of other markers) will not capture this evolution. As discussed below, a more thorough transcriptomic analysis might help identify if there are a more diverse set of cells in these cultures.

There is high enthusiasm about analyzing human mesenchymal development as a means to identify factors that play a role in endocrine cell development. However there were several areas of weakness in the transcriptomic approach. First, one premise of the paper is that the "niche" changes during developmental time and that that these changes are what dictate the progression of progenitors to mature beta cells. However in the transcriptomics experiments HDFs were used as comparator (negative control) to wk 17.5 and 20. This makes little sense since HDFs (skin fibroblasts) may bear little resemblance to anything in a human fetal pancreas and any differences may not be due to "niche" factors. A better comparison would be with wk9.1 H-E lines, since they were also inefficient in promoting ins+ cells and thus should be lacking key inducing factors. In addition to adding wk9.1 to the data set, a more comprehensive bioinformatic analysis should be performed to show broad changes in mesoderm cell types, signaling pathways, ECM etc. over time. It is possible to tease out mesenchymal cell populations even from bulk RNAseq data, in part by comparison with other data sets.

With regards to identifying factors that may control Ins+ cell development, at the moment the manuscript shows a highly selected group of transcripts that are changed relative to skin fibroblasts, and there may be no correlation with the ability of subpopulations of pancreatic mesenchyme to induce endocrine cell fate. For example, it could be possible that Wnt5 is made throughout these stages of embryonic mesenchyme development (9-20wk) and it acts in a trophic manner to enhance endocrine cell formation. Conversely it may not be present in 9wk cultures but come on at later stages.

While the M-E cultures appear to work better than HUVECs or MEFs to induce ins+ cells, one obvious possibility is that HUVECs plus MEFs would have more substantial endocrine maturation effects. Did the authors test this? Work done in Takebe et al (Nature 2013) shows that MSCs plus HUVECs can have a massive maturation effect on iPSC-derived hepatocytes.

M-E conditioned media seems to promote the formation of ins+ cells starting from both PP and EP cells. This would suggest that the factors made by M-E promote the PP to EP transition as well as the maturation into insulin secreting cells. Since KGF, BMP inhibition, TGFb inhibition, smoothed inhibition all are used to promote PP to EP transition, did the authors find regulatory factors that modulate these pathways in their M-E transcriptional profile? The GO terms suggest that the TGF-beta pathway is increased, which is counterintuitive since an ALK5inhibitor is used to promote EP development. Some more thoughtful analysis and discussion of the data in the contexts of pancreas development would be helpful.

Along those lines, it is clear from embryonic development and from figure 7 that inhibition of pathways may be just as critical as stimulation of pathways. BMP inhibition may be one such example, as the authors have shown that Wnt5a treatment results in reduced BMP signaling. However it is not clear if the endogenous BMP signaling pathway is regulated in the M-E cultures. One might expect that BMP signaling is high in early cultures but down in wk 20 M-E cultures. Are BMP ligands down in 20 wk cultures or conversely are BMP antagonists up?

Wnt5a acts through the non-canonical Wnt/PCP pathway. The role of Wnt/PCP pathway in islet cell development and maturation has been previously demonstrated in mice using the Wnt/PCP pathway effector Flattop (Bader et al, 2016 Nature). In this context, namely during islet formation and beta cell maturation in mice, the Wnt/PCP pathway triggered beta cell maturation. This was also seen in human

beta cell lines and human micro islets. Do human PP and EP cells express Flattop in response to Wnt5? Is the maturation seen in this work the same effects as seen in the Bader paper? This point should be acknowledged and discussed in more depth. Currently the authors state "recent studies showed that Wnt5A induces proliferation of some β cells (Bader et al., 2016)." This is both inaccurate and is glossing over a seminal paper that is the first to show Wnt5a regulates beta cell maturation in the early postnatal mouse (a stage of development that is likely similar to the EP maturation being modeled in this manuscript).

The authors completely misuse the term niche. The term niche is meant to represent a defined structure that exists in a living organism which through its structure and production of factors is able to regulate stem/progenitor cell maintenance and differentiation. It can be made up of many cells that through their orientation and production of secreted factors and ECM, and cell-cell interactions can control stem and progenitor cell replication and differentiation. While mesenchyme and vascular cells in culture may provide factors in vitro that can promote maturation of HESC-derived PPs, they are not a niche. The authors should remove the term niche from the manuscript unless they plan to show a more extensive analysis of human embryonic/fetal pancreas, by imaging all the cell populations and their relationship to the developing endocrine pancreas. This could be done by single cell RNAseq of fresh pancreas followed by 3d imaging of individual cell populations to show both the spatial and temporal nature of the "niche" relative to developing endocrine progenitors. It has taken decades of work from many labs to come up with a model of what makes up the niche for the stem cells of other organs (germ line, blood, skin, intestine), and these studies are far from complete. In this study, the extent of the "niche" studies are that Wnt5 protein staining is observed in a section from a 16week fetal pancreas.

It is very interesting that E-M matrix appears to promote ins⁺ cells in a manner similar to conditioned media (Fig 2e). However this point wasn't discussed further. Do the authors have any insight into matrix effects on PP to EP to beta cells? Perhaps this could be discussed following a more thorough analysis of the RNA seq data.

Endocan is inducing a substantial differentiation into endocrine cells as measured by CHGA, however these are not Ins⁺ cells. What is endocan doing? This is a very interesting result and is perhaps one of the most novel findings to come from this approach. Does this promote a pan-endocrine cell? Is it another lineage?

The experiment where Wnt5a-treated EPs were engrafted into a mouse is entirely preliminary, is not conclusive, and adds little to the manuscript. N=1 mouse, one section with Ins and Pdx1 staining does not merit the conclusion that wnt5a treated EPs "differentiate into beta cells" or that they are "able to respond to glucose stimulation in vivo". However given the nice set of in vitro experiments showing GSIS in M-E co-cultures (Fig 2A-A'''), it would stand to reason that the authors would have attempted to show that cultures treated with Wnt5a + Gremlin show improved GSIS. If not, then that would suggest that Wnt5a/BMPi is involved in ins⁺ cell development but not maturation.

Minor comments

Figure 2A and A' seem to be redundant with A'' and can be moved to supplement.

Figure 3 is all transcript based so the descriptions need to reflect this. The title in the results suggests that growth factor and ECM proteins were analyzed.

We thank the Reviewers for all their comments and overall positive assessment of our work. Following the suggestions, we have included new data and significantly reorganized and rewritten parts of MS that collectively, in our opinion, improved the scientific strength and clarity of this work. The detailed response to each comment is included below, in blue font. Thank you again for your time and input.

Reviewers' comments:

Reviewer #1 (Remarks to the Author):

In the paper “Human pancreatic niche promotes β cell differentiation via WNT5A/JNK/AP1 and BMP signalling” Yang et al describe the function of Wnt5a in pancreatic islet development and in particular in Insulin positive β -cells. In a screening attempt, the authors identify Wnt5a as a crucial factor guiding pancreatic progenitor cells to β cells. The authors demonstrate that Wnt5a is important for the pancreatic cell differentiation namely from pancreatic progenitors to endocrine progenitors to β cells. Furthermore, the authors suggest that Wnt5a activates the JNK pathway. Finally the authors suggest that Wnt5a/JNK signalling inhibits the BMP pathway in a later stage of development - especially during β cell maturation. In general the author use a wide variety of techniques to support their conclusion. However, I have some difficulties with this work: First, I feel that the function of Wnt5a signalling in pancreatic development is not a novel finding. A detailed study from Kim et al., 2005 analysed the function of Wnt5a in pancreas development already in zebrafish, Xenopus and mouse. At the end of the work, Kim et al came to the conclusion that Wnt5a is important for β cell migration and therefore for niche formation. For example Kim et al finds no reduction of insulin positive β cells in the zebrafish knock-down and in the Wnt5a ko mouse. In the work presented in this manuscript, the authors believe that Wnt5a is important for the induction of the differentiation programme of the pancreatic niche in particular the β cells, which is fundamentally different to the conclusion of Kim et al. Unfortunately, the authors do not discuss this discrepancy. Therefore, it is unclear if the different results can be explained by a human specific function of Wnt5a in comparison to other vertebrates. A further explanation of the difference would be based on the experimental system: Kim et al used an in vivo approach (overexpression, knock-down and knock-out in vertebrate model organisms) whereas Yang et al uses an in vitro cell culture approach with human stem cells. The author should address this discrepancy. A possibility to address it would be to add convincing human organoid studies - the xenograft model is insufficient to address this problem.

Indeed, Kim et al. 2005 work elegantly shows the necessity of Wnt5a for β -cell migration but not β -cell formation. However, in the work of Larsen et al. 2015, Wnt5a rescued the β -cell formation in a *Hox6b* knock-out murine model. Mesenchyme-derived *Hox6* knock-out was found to completely abolish Ngn3+ cells progression to endocrine cells. *Wnt5a* expression was found to be strongly inhibited in the KO's mesenchyme (but not in epithelium) and addition of recombinant Wnt5a rescued endocrine cell formation in e12.5 explants, showing the pro-endocrine role of Wnt5a. It is thus possible that Wnt5a acts both on endocrine cell

induction (which is our focus) and migration (which we didn't elaborate here). The discrepancy might also result from redundancy Wnt5a and Wnt5b within the endocrine specification. In Kim et al. study Wnt5b gene was not disrupted, while in the *Hox6* KO (Larsen et al.) Wnt5b expression is downregulated (based on the Larsen microarray dataset - <https://www.ncbi.nlm.nih.gov/geo/geo2r/?acc=GSE68390>). We now discuss this thoroughly in the MS.

Here we evaluated human mesenchyme-expressed WNT5A and identified WNT5A downstream effectors (JNK and BMPs). Addition of WNT5A and GREM1 to the human PSC endocrine differentiation protocols improved the efficiency of β -cell differentiation. Furthermore, in the revised MS, we also included data on Endocan, which is the first identified recombinant factor shown to induce SST+ expressing δ -cell from hPSCs.

Detailed comments:

Fig2D M-E supernatant and ECM enhance β cell differentiation. The authors need to provide additional data for this claim as the experimental system does not allow this rigorous conclusion.

Following the Reviewer's suggestion, we have extended this part as follows. We have added Collagen IV and Laminin immunofluorescence staining (Fig. S4), to show that our decellularization method worked, leaving only intact ECM on a plate. We have also added quantification for both ECM and CM effects on C-peptide+ cells induction (this is now Fig. 2B-E) and we have included additional controls like heat-inactivated CM.

Fig. 2F Change orientation of the graphs

We have merged and changed the orientation of the graphs (this is now Fig. 2G).

Fig 3CDE: Explain colours used in the presentation

The microarray data in Fig. 3 have been replaced with RNA-Sequencing data. The existing heatmap (Fig. 3D) contains a legend depicting the colour scale of log₂ Z-score values.

Fig. 3 The gene expression array data should be backed up by a deep sequencing analysis.

Following the Reviewer's suggestion, also pointed out by Reviewer # 3, we have performed the RNA-sequencing of M-E cells, (from Wk9, Wk17.5 and Wk20.1) control cells (HDFs, HUVECs) and hPSC-derived EPs. Importantly, the analysis of RNA-sequencing from the independent cultures of M-E and control cell, is consistent with the microarray data analysis. In the revised manuscript we have replaced the microarray data with the RNA-Sequencing data (Fig. 3), for the presentation clarity. Furthermore, we have extended the transcriptome analysis and performed the ligand-receptor interactome analysis between hPSC-derived EPs and M-E cells. To this end we cross-referenced our RNA-sequencing data with the FANTOM5 database and performed the ligand-receptor analysis, which coupled ligands expressed in M-E cells with their receptors expressed in EP cells. This analysis pointed out that several candidate growth factors, which we used for further evaluation (e.g., WNT5A, LIF, THBS2, FGF7, HGF), have their known receptors expressed in EP cells.

Fig. S4 B-C The pictures do not show an increase of INS β cells after Wnt treatment. Therefore, it is difficult to believe the statistics if these are the best pictures the authors can provide.

Not to mislead the statistical conclusion based on the pictures we made the new Fig. S7A-B. Thank you for letting us know that these pictures were not representative. We provided IF staining for multiple growth factor treatments and included picture for WNT5A only treatment, as well as updated the representative images for growth factor treatments.

Fig5 A The co-localisation of Wnt5a and Pecam or Vimentin is not convincing. The authors have to provide better data.

Thank you for letting us know that these images were not convincing. We have replaced the VIM/WNT5A co-staining image and added insets with enlarged image for better visualization. Furthermore, we now included the quantification of colocalization, see below.

Fig 5B Furthermore a quantification is required illustrating the cells expressing both markers and cells expressing only one marker.

We have replaced the qPCR results with quantification of WNT5A+ cells within mesenchymal (VIM+) or endothelial (PECAM+) compartments (Fig. 5B). The quantification showed that 52% (95% CI: 23-82%) of VIM+ cells and 33% (95% CI: 24-42%) of PECAM+ cells express WNT5A.

We have all added the qPCR data for M-E expression of the tested growth factors in Fig. S5.

Fig 5E I see a 50% reduction after Wnt5a ko however the author write “a 3.6fold fewer cells”.

We have replaced the data for M-E17 with more comprehensive M-E20 KO experiments (now Fig. 5F), including the missing control coculture (for which the value was given). We have included results for KO cells derived using different sets of sgRNA, as well as WNT5A rescue experiments. We also added the representative images for WNT5A-KO coculture in 3D differentiation assay (Fig. 5G).

Figure 5E The authors have to provide a rescue experiment by adding Wnt5a to the medium or by using Wnt5a transfected Wnt5a ko cells as control.

We thank the Reviewer for this suggestion. The rescue experiments for M-E WNT5A KO coculture has been included in the graph (which is now Fig. 5F). In these experiments we added WNT5A to the culture and showed a partial rescue with a significant 2.5-3-fold increase in the number of INS+ cells as compared to M-E WNT5A KO coculture.

Fig 5F is not sufficient as a control experiment as it is technically different.

To keep it technically consistent, the data for EPs coculture with HDFs overexpressing WNT5A has been removed.

Fig S5D Is Wnt5a expressed in the same cells as Fzd3. How do the authors explain the results from Kim et al 2005 suggesting Fzd2 is receptor of Wnt5a in other vertebrate models?

According to our data WNT5A is mostly expressed in M-E cells, while it acts in FZD3-expressing EPs. It is possible that WNT5A can act through various receptors depending on the context and function. Kim et al. described Fzd2 as a Wnt5a receptor expressed in migrating β -cells, which may require different mechanisms. We found FZD3 as the WNT5A receptor present in human EPs (Fig. 3 ligand-receptor analysis of RNA-Sequencing, Fig. S10B immunofluorescence) and proved its functional role in β -cell induction by anti-FZD3 treatment (Fig. S10C), which also is in an agreement with other work (Larsen et al., 2015).

Figure 5G-H shows no dose dependency. The authors should provide alternative data.

We included two concentrations of plasmid used for WNT5A OE (Now: Fig. 5C-D, Fig. S8A), which shows a significant dose-dependent INS induction.

Figure S6A The author should provide an alternative experiment to support their claim that there is no effect on proliferation but on differentiation.

Following the Reviewer's suggestion, we have included additional data on proliferation assay of EPs after WNT5A treatment. Together, we showed no significant WNT5A induction on proliferation (Fig. S9). In details: despite the initial increase of INS⁺ cells after 4 days, after 12-day EP culture with WNT5A, the number of INS⁺ cells is comparable to control (no WNT5A), which suggests that β -cells are induced from the existing EPs, without stimulation of β -cell proliferation (Fig. S9A). We also did proliferation quantitative assessment after WNT5A treatment of EPs by pH3 staining and EdU incorporation assay (Fig. S9B).

The cell migration experiments (Boyden, scratch) are not clear to me and needs a more convincing demonstration. Furthermore, the author should provide a positive controls in these experiments.

We have performed multiple repeats of the migration assay experiments and we did not observe any significant effect imposed by WNT5A treatment on EP migration. Having said that, we agree with the Reviewer's comment about positive control - although it is not obvious which growth factor could serve as such control in our system. Due to this and to focus the MS on a single message we decided to remove the cell migration experiments from the MS.

Furthermore, it looks to me that the cells start to form cluster after Wnt5A treatment e.g. in Fig. 4F, 5G which do point towards a Wnt5A dependent migration effect.

Based on our observations endocrine cells often form clusters in 3D pancreatic differentiation, also seen in a representative image of control (now Fig. S9A). Similar observations have been reported by others (Sharon et al., Cell, 2019) and it is proposed the "peninsula" model for islet morphogenesis. In addition, during *in vitro* pancreatic differentiation, the mesenchyme is mostly absent, thus delaminating EPs and early endocrine cells cannot migrate through the non-epithelial microenvironment. Therefore, we conclude that the clustering is not an effect of WNT5A treatment but rather a general EP phenomenon during *in vitro* differentiation. We apologize for not selecting more representative images in the previous manuscript.

Line 388 : Genes MAFB, NGN3, What is the reason behind the selection of these markers and an interpretation of this result is missing.

We have revised this part of MS and the corresponding Fig. 6B.

Fig 6E,F Could the authors add Wnt5A to overcome the effect of the JNK inhibitor? Can the effect of Wnt5a dependent stimulation of INS positive cells reverted by blocking JNK signalling? These combinatorial experiments are crucial to demonstrate that specifically Wnt5a activates the generic JNK pathway.

Thank you for this important suggestion. The experiments with JNK inhibitor along with WNT5A have now been added (Fig. 6I), showing that the JNK inhibitor is sufficient to overcome WNT5A stimulation and that WNT5A cannot surpass the inhibitory action of JNK_i. Together it shows that WNT5A promotes β -cell induction via downstream JNK.

Fig 6S A Is the increase of pH3 cells from 1% to 2% or is the dataset normalized and this represents a relative increase from 100%? The picture in A suggests rather the second option.

The images have been changed to state that the data was not normalized but reflected the „1% to 2%“ change (now: Fig. S9C).

Fig S6 C Scratch assay: Very difficult to see - quality of the picture needs improvement. How do the authors explain that Wnt5a treatment (highest concentration) has an opposite effect regarding cell migration? The author should provide a positive control.

Fig S6 D Boyden chamber: The author should explain why 500ng/ml of Wnt5a show a lower activity compared to Wk17.5 conditional medium. How do the authors explain the difference? The author should provide better controls.

We decided to remove these results from the MS, according to reasons described above and the other Reviewers' recommendations.

Fig S7 It is unclear to me when the xenografts have been treated with Wnt5a. I suspect that the cells have been treated with Wnt5a prior implantation. Therefore the starting material would be different. Unfortunately it would be an in vitro expansion of the β cells and an in vivo confirmation of the maintenance of the cell fate. The title and conclusion are mis-leading.

We agree that such experiments are quite complex because of the variability of both the starting materials and the host *in vivo* environment. Furthermore, the xenograft experimental conditions remain vague with many potential signaling events occurring *in vivo*. Therefore, the interpretation of such results might be difficult, and it requires consideration of different possibilities, including these brought up by the Reviewer. Therefore, we have removed these results from the revised manuscript.

General comments: Although, the small schematics in Fig. 2B,C,F are helpful, the order and the presentation of the data in this manuscript needs to be revised. In

general, figures needs a better description, sometimes higher-magnification of particular areas would be advantageous and some importantly - some sets of experiments are hidden in the supplementary data. The readability of the text needs to be further improved.

Thank you for these suggestions. The manuscript and figures have been carefully and thoroughly revised as the Reviewer suggested, which in our opinion, improved the clarity of the manuscript.

--

Reviewer #2 (Remarks to the Author):

There is something positive happening with the main claims of the paper, in that (I think) the "in vivo" study of the secreted intercellular signaling proteins coming from the human pancreatic niche (non-endodermal cells) can yield candidate proteins that increase the number of well-differentiated beta cells (insulin-producing) in in vitro protocols for the differentiation of such from the pluripotent ES or iPS lines. A problem with the m/s is that the level of care in preparing it for submission seems below par. I have annotated the m/s with a large number of notes all over the text and the figures that are designed to help. There are some real problems with some of the data, and complete inconsistencies between the main text and the figures/figure legends. Part of the problem could be the number and density of information presented - it is a huge paper after all. I do feel that a significant amount of the data can be either removed as unnecessary - OR, the reason for its inclusion must be added. Again, some of my notes indicate these areas. The main theme and significance of the findings remain non-explicit still, in this version.

Stylistically, there are some areas where the language and grammar leave a great deal to be desired. I tried to catch some/most of these, but there are likely others. Sometimes they are quite annoying to have to read through.

My recommendation is that the authors be given a chance to look over this set of comments from me (I have not transposed them all into this box, but left them on the annotated m/s), and then prepare a much better m/s that should in some ways only then be regarded as the "first" submission. I think there are some interesting findings that are suitable for Nat Comms., but it needs some serious reworking before being able to work through it in a more full-fledged review.

We appreciate the Reviewer's comment regarding our language issues, which made it impossible to recognize our findings. We ameliorated the text according to the Reviewer's explicit suggestions and hope the Reviewer will find it now acceptable. To improve the flow, we have rewritten the text and presented data in a coherent way. Some of the data has been removed (e.g., WNT5A overexpression in HDFs, the scratch assay, transwell and xenotransplantation experiments) as the Reviewer suggested or moved to supplements (e.g., GSIS data), while some relevant experiments were extended or added (e.g., M-E RNA-Sequencing in Fig. 3, Endocan in Fig. 4, and GSIS after WNT5A+Grem1 treatment in Fig. 7).

Specific issues raised by the Reviewer:

Line 120-121: incorrect to claim this - how are they in any way "mature" given the current discussion of only post-natal maturation of islet cell types?

We agree with the reviewer that the sentence was misleading, and it has been removed. This part of manuscript (the first paragraph of Results) has been thoroughly rewritten.

Lines 211-212: this result means WHAT, essentially? A higher level interpretation is needed, related to the theme of the paper.

According to the suggestion, we have extended this description of ECM and conditioned medium (CM) experiments, and the results, as below:

Lines 189-95: "However, the efficiencies were lower than in the coculture experiments (Fig. 1E and 2D), pointing to complexity of M-E mediated pro-endocrine function.

Despite the higher efficacy of β -cell differentiation in cocultures, this method has a limited potential for regenerative medicine. Knowing that CM from specific M-E cells significantly promotes PP maturation into INS+ cells, we focused on defining the M-E secreted factors, which can be easily applied into the culture medium during *in vitro* differentiation."

Lines 304-316: "Were all 4 tested together, WNT5A plus the others?"

Yes, all four candidate growth factors were tested together, and these results are now included in Fig. S7A-C. Combinatorial treatment of four growth factors did not cause a higher induction of INS+ cells than treatment with WNT5A only or group of three growth factors, except for those with HGF. For HGF, we saw a negative effect on INS+ cell induction.

Lines 304-316: Concerns that use of human ovarian cell line staining does not prove specificity of WNT5A antibody, and that Wnt5a-null cells or other control is needed.

We agree with the Reviewer, and we have removed this data. We have generated WNT5A-KO M-E20 (mesenchyme-endothelium Wk20.1) cells and now included IF staining of M-E20 WT and WNT5A-KO cells (Fig. 5E), which clearly proves the WNT5A antibody specificity.

Lines 332-333: Concerns about description of result shown in Fig. 5E and discrepancy with the Fig. legend.

We have included missing M-E20 coculture conditions and added the rescue experiment with WNT5A addition (this is now Fig. 5F). We also corrected labelling, the figure legend and rewrote the experiment description in the manuscript.

Now it reads: "Coculture of EPs with M-E20 W5A KO cells yielded 9-fold fewer INS+ cells than coculture with control (WT) M-E20 cells (Fig. 5F). Further, by adding WNT5A to the coculture of EPs and W5A KO, we partially rescued the INS+ cell induction (Fig. 5F)."

Line 342-345: about WNT5A overexpression and antibody blocking experiments. "there is NO point in these additional experiments, is there? is it different from the Wnt5A addition experiments, conceptually?"

We think that showing WNT5A effects with different methods strengthens our conclusions, and that gives some additional information, like in case of antibody blocking experiment. However, we revised description of all experiments performed to prove WNT5A role to make this part of manuscript more concise.

Fig. 1A. all of these: cell lines (clonal) or cells? if not clonal in origin, they are not formally cell lines.

Thank you for pointing this out. We have changed the nomenclature from “cell lines” to “primary cells” in the Fig. 1A and through the MS text.

Fig 1F. NOT a whisker plot, nor is 2E; 2F and other places do show box- and-whisker plots

Thank you for pointing this out. We have changed legend of the current Fig. 1E and 2E to the correct “scatter plot”.

Fig 2A. what's this "u" should be micro sign (greek mu); what is A and A' compared to A"? not explained in text - different experiments? cannot understand; KCl in text (see legend) but is irrelevant here as not shown. Care in prepn. of m/s needed.

Thank you for pointing out that this figure was unclear. We have corrected “u” with the micro sign “ μ ”, corrected figure legend and captions, as well as exchanged box plots to scatter plots. Fig. 2A” (currently Fig. 1G) summarizes GSIS results after 2.8 mM and 16.7 mM glucose treatments for d3, d7 and d14 of coculture. Former Fig. A and A’ have been moved to supplementary Fig. S3. Current Fig. S3B-D consists of three scatter plots corresponding to Fig. 1G. Each plot summarizes a separate test day (3, 7, or 14) for both 2.8 mM glucose, 16.7 mM glucose and KCl treatment. The corresponding manuscript part has been thoroughly revised for flow and clarity.

Fig 2C. conditional?

We apologise for this mistake. We have corrected it to “conditioned” in figure legend, while in the figure it is currently abbreviated to CM. This is Fig. 2A now.

Fig 2E. $p < 0.05$ inappropriate

Use exact values for p values in all places - it is the only way to do it!.

We have now included the exact values for p values in all relevant figures.

Fig 4E. is this a percentage? explain!

Axis labels in current Figs. 4d-f have been corrected, and now include [%]. We apologize for this oversight.

Fig 4G. so what's going on here? why 12 days gives essentially the same percentage is not explained or discussed in the text

This figure (and former Fig. 4F) has been moved to supplements, and they are now Figs. S9A-B. We have now explained this in the manuscript text, as follows:

"We noted that the prolonged (12-day) treatment with WNT5A of EP-stage spheres does not result in more INS+ cells, as compared to control based on Pagliuca et al. protocol (Figure S9A-B), suggesting that WNT5A accelerates β -cell differentiation of the existing EP pool."

This is followed by the extended experiments confirming lack of WNT5A influence on proliferation rate.

5A. cannot work out anything from this (5A) - need to show epithelium not making it; lower power, too; different presentation to make the point. Nothing is clear here.

We have exchanged the image on the right, as well as added magnification insets. We have also included quantification of this data (Fig. 5A), which shows the fractions of endothelial (PECAM+) and mesenchymal (VIM+) in which WNT5A is expressed.

5B. control for FZD3 Ab?

We have obtained the FZD3 Ab from Dr. Jeremy Nathans, who is an expert in Frizzled protein biology, and who had validated this antibody in his works on FZD3 knock-out mice. The figure has been moved to supplements (Fig. S10B).

5D. too small, too few cells

The M-E WNT5A KO image has been changed (it is now Fig. 5E). The blot has been moved to Fig. S8D.

5E. CONTROL is no coculture here. Should include Wk17.5h(no KO)

We have included the WT M-E (no KO) coculture control. Moreover, we extended experiments with additional KO derived with additional sgRNA pair and by rescue experiments with recombinant WNT5A, which further support our observations, as well as by experiments in 3D suspension format. The extended experiments are based on M-E20 WNT5A KO and are shown as Fig. 5E-G.

5G. NOT impressive, or control is misleading

We have exchanged the image showing WNT5A expression for EPs transfected with control plasmid for a more representative one (which is now Fig. 5C).

Reviewer #3 (Remarks to the Author):

Summary

This manuscript describes the use of human fetal pancreatic explants to generate primary mesenchymal and endothelial cultures, which were then used to promote differentiation of human embryonic stem cell-derived pancreatic and endocrine progenitors. The mesenchymal cells in the cultures were validated with vimentin staining and endothelial cells by PECAM1 and CFIII. The authors noted that these are primary cultures that change over time, thus all experiments were done with cells

within the first 6 passages. M-E cultures were used to promote the differentiation of pancreatic progenitors (PP) into insulin + cells, and were shown to be more effective than control fibroblasts. Interestingly M-E cultures from 9 week were not very effective in promoting ins+ cell production whereas 17 and 20 week M-E cells produced factors (both ECM and soluble) that enhanced both insulin expression and GSIS. The authors performed RNAseq on 17 and 20 week M-E cultures to identify candidate factors that may promote PP and EP maturation, and several of these factors were tested for their ability to enhance differentiation, in particular Wnt5a. Wnt5a is a non-canonical wnt/PCP ligand was able to substantially enhance production of ins+ cells, consistent with its role in mouse beta cell development/maturation. Wnt5a was genetically ablated from 17 week M-E cultures and was shown to be necessary for full ins-inducing activity. Wnt5a-treated cultures had reduced expression of BMP ligands and increased BMP antagonists, suggesting that repression of BMP signaling may be important for ins+ cell production. Indeed inhibition of BMP with Gremlin promoted ins+ cell numbers.

The main findings, namely the isolation and characterization of human fetal M-E cultures and their use to promote the differentiation of PP and EPs into ins+ cells, the identification and functional studies of Wnt5a in promoting differentiation of ins+ cells, are all interesting, important and very well supported by the data. There are quite a few examples where the language used and the conclusions reached are not supported by the data. There are also several examples where additional experiments would strengthen the conclusions and manuscript in general. These are all discussed in detail below.

Comments

While there is tremendous excitement over the premise of identifying embryonic populations of “niche” cells and studying how they change over time, this manuscript in its essence does not do this. The mere act of transferring cells to plastic and growing them as monolayers for months with multiple passages would dramatically change cell composition through selection. The minced tissues were grown in DMEM:F12 with 10%FBS, which will allow for robust expansion of fibroblast-type cells, but may select against many other mesenchymal populations. The end result is that the heterogeneous population of cells at the start is likely to have a more homogeneous population of cells after months of passaging. Analysis of Vimentin and PECAM (and a handful of other markers) will not capture this evolution. As discussed below, a more thorough transcriptomic analysis might help identify if there are a more diverse set of cells in these cultures.

There is high enthusiasm about analyzing human mesenchymal development as a means to identify factors that play a role in endocrine cell development. However there were several areas of weakness in the transcriptomic approach. First, one premise of the paper is that the “niche” changes during developmental time and that that these changes are what dictate the progression of progenitors to mature beta cells. However in the transcriptomics experiments HDFs were used as comparator (negative control) to wk 17.5 and 20. This makes little sense since HDFs (skin fibroblasts) may bear little resemblance to anything in a human fetal pancreas and any differences may not be due to “niche” factors. A better comparison would be with wk9.1 H-E lines, since they were also inefficient in promoting ins+ cells and thus should be lacking key inducing factors. In addition to adding wk9.1 to the data set, a more comprehensive

bioinformatic analysis should be performed to show broad changes in mesoderm cell types, signaling pathways, ECM etc. over time. It is possible to tease out mesenchymal cell populations even from bulk RNAseq data, in part by comparison with other data sets.

Thank you for bringing up these important suggestions. Following your suggestion we replaced microarray with RNA-sequencing analysis and included M-E9 cells as additional control to HDFs and HUVECs. We have also performed RNA-sequencing of EPs to perform ligand-receptor analysis and identified M-E secreted ligands that have their receptors expressed in EPs (Fig. 3E, F).

With regards to identifying factors that may control Ins⁺ cell development, at the moment the manuscript shows a highly selected group of transcripts that are changed relative to skin fibroblasts, and there may be no correlation with the ability of subpopulations of pancreatic mesenchyme to induce endocrine cell fate. For example, it could be possible that Wnt5 is made throughout these stages of embryonic mesenchyme development (9-20wk) and it acts in a trophic manner to enhance endocrine cell formation. Conversely it may not be present in 9wk cultures but come on at later stages.

We can appreciate what the Reviewer is pointing out; we performed follow-up for a highly selective group of soluble factors. The transcriptomic experiments (both former microarrays and current RNA-sequencing analysis yield consistent datasets) allowed us, combined with an extensive literature search, to select candidate factors for further analysis. Therefore, we did not include in follow-up experiments all M-E derived factors. We think that differences observed between M-E cells derived at different stages in their ability to induce β -cell fate cannot be simply explained by expression of a single factor but rather by the collective action of all soluble and ECM factors that can be received and processed by progenitors. Indeed, the coculture effect on EP maturation is still stronger than WNT5A treatment alone, as an example regarding GSIS. To evaluate such complex interactions, it would require a systems biology approach that was not intended for this project.

However, in our opinion, this does not contradict the important role of the identified factors in human endocrine cell specification, namely WNT5A and mechanisms involved for β -cells and Endocan for δ -cells.

While the M-E cultures appear to work better than HUVECs or MEFs to induce ins⁺ cells, one obvious possibility is that HUVECs plus MEFs would have more substantial endocrine maturation effects. Did the authors test this? Work done in Takebe et al (Nature 2013) shows that MSCs plus HUVECs can have a massive maturation effect on iPSC-derived hepatocytes.

We have included results for coculture with MEF along with HUVECs (Fig. 1E), which was surpassing INS⁺ induction caused by Wk9 (M-E9) coculture, but it was less effective than Wk17.5 (M-E17) and Wk20.1 (M-E20).

M-E conditioned media seems to promote the formation of ins⁺ cells starting from both PP and EP cells. This would suggest that the factors made by M-E promote the PP to EP transition as well as the maturation into insulin secreting cells. Since KGF,

BMP inhibition, TGF β inhibition, smoothed inhibition all are used to promote PP to EP transition, did the authors find regulatory factors that modulate these pathways in their M-E transcriptional profile? The GO terms suggest that the TGF-beta pathway is increased, which is counterintuitive since an ALK5 inhibitor is used to promote EP development. Some more thoughtful analysis and discussion of the data in the contexts of pancreas development would be helpful.

Thank you for this interesting comment. We agree with the Reviewer's suggestion that multiple signaling pathways control the PP to EP transition and therefore perturbants of several signaling pathways, including inhibitors of BMP, TGF β and KGF, are included in common pancreatic differentiation protocol to coax PPs into becoming EPs. We cannot exclude some cross-regulatory interactions between these different signaling pathways during pancreatic specification and thus some of these pathways might be redundant in their effect on EP formation *in vitro*. When we co-cultured PPs with M-E cells we were able to obtain INS+ cells without addition of these perturbants. Interestingly, we found KGF (FGF7), BMP inhibitors (see below), Smad6 which is a TGF β inhibitor (Fig. S11) upregulated in M-E17 or 20 suggesting that these cells might be a sufficient source of these signals critical for EP formation.

Along those lines, it is clear from embryonic development and from figure 7 that inhibition of pathways may be just as critical as stimulation of pathways. BMP inhibition may be one such example, as the authors have shown that Wnt5a treatment results in reduced BMP signaling. However it is not clear if the endogenous BMP signaling pathway is regulated in the M-E cultures. One might expect that BMP signaling is high in early cultures but down in wk 20 M-E cultures. Are BMP ligands down in 20 wk cultures or conversely are BMP antagonists up?

Indeed, the BMP antagonists *BAMBI*, *BMPER* and *DCN* are enriched in M-E17 and M-E20 as compared to M-E9, which is now presented as a heatmap in Fig. S11.

Wnt5a acts through the non-canonical Wnt/PCP pathway. The role of Wnt/PCP pathway in islet cell development and maturation has been previously demonstrated in mice using the Wnt/PCP pathway effector Flt1p (Bader et al, 2016 Nature). In this context, namely during islet formation and beta cell maturation in mice, the Wnt/PCP pathway triggered beta cell maturation. This was also seen in human beta cell lines and human micro islets. Do human PP and EP cells express Flt1p in response to Wnt5? Is the maturation seen in this work the same effects as seen in the Bader paper? This point should be acknowledged and discussed in more depth. Currently the authors state "recent studies showed that Wnt5A induces proliferation of some β cells (Bader et al., 2016)." This is both inaccurate and is glossing over a seminal paper that is the first to show Wnt5a regulates beta cell maturation in the early postnatal mouse (a stage of development that is likely similar to the EP maturation being modeled in this manuscript).

We did not observe Flt1p mRNA induction upon 12h or 5d WNT5A treatment of EPs by RNA-Sequencing and qPCR. In the Bader et al. work, Flt1p was induced in β -cells that were losing proliferative capacity and undergoing morphological and functional maturation within endocrine mouse and human islets and EndoC β -H1 cell clusters. Importantly, the authors found that Flt1p was not necessary for β -cell development. In our work we focused *per se* on

the role of WNT5A in β -cell fate commitment for which the downstream Fltp pathway is likely not involved. It is possible that Fltp expression could be induced upon WNT5A treatment during the last stage of differentiation and improves functional maturation of β -cell, which we have not tested as in this work.

The authors completely misuse the term niche. The term niche is meant to represent a defined structure that exists in a living organism which through its structure and production of factors is able to regulate stem/progenitor cell maintenance and differentiation. It can be made up of many cells that through their orientation and production of secreted factors and ECM, and cell-cell interactions can control stem and progenitor cell replication and differentiation. While mesenchyme and vascular cells in culture may provide factors in vitro that can promote maturation of HESC-derived PPs, they are not a niche. The authors should remove the term niche from the manuscript unless they plan to show a more extensive analysis of human embryonic/fetal pancreas, by imaging all the cell populations and their relationship to the developing endocrine pancreas. This could be done by single cell RNAseq of fresh pancreas followed by 3d imaging of individual cell populations to show both the spatial and temporal nature of the “niche” relative to developing endocrine progenitors. It has taken decades of work from many labs to come up with a model of what makes up the niche for the stem cells of other organs (germ line, blood, skin, intestine), and these studies are far from complete. In this study, the extent of the “niche” studies are that Wnt5 protein staining is observed in a section from a 16week fetal pancreas.

We agree with the Reviewer that the niche is a complex, organized entity and that the dispersed niche-derived cells cultured *in vitro* cannot reflect this complexity. Therefore, we have removed the term niche from the manuscript when used in the context of the derived mesenchymal-epithelial cells, and instead named these niche-derived cells “M-E cells”.

It is very interesting that E-M matrix appears to promote ins⁺ cells in a manner similar to conditioned media (Fig 2e). However this point wasn't discussed further. Do the authors have any insight into matrix effects on PP to EP to beta cells? Perhaps this could be discussed following a more thorough analysis of the RNA seq data.

We have added quantification of the ECM effects in Fig. 2C, but to keep this work focused on the soluble factors and eventually WNT5A, we decided not to explore the ECM role more extensively within this work.

Endocan is inducing a substantial differentiation into endocrine cells as measured by CHGA, however these are not Ins⁺ cells. What is endocan doing? This is a very interesting result and is perhaps one of the most novel findings to come from this approach. Does this promote a pan-endocrine cell? Is it another lineage?

Indeed, results for Endocan are very exciting. We have now included results (Fig. 4G-H and Fig. S5E) showing that Endocan induces SST⁺ δ -like cells, which is the first identified growth factor to do so. We have a project in progress, which further explores this topic, and thus prefer not to extend this data here.

The experiment where Wnt5a-treated EPs were engrafted into a mouse is entirely preliminary, is not conclusive, and adds little to the manuscript. N=1 mouse, one

section with Ins and Pdx1 staining does not merit the conclusion that wnt5a treated EPs “differentiate into beta cells” or that they are “able to respond to glucose stimulation in vivo”. However given the nice set of in vitro experiments showing GSIS in M-E co-cultures (Fig 2A-A’”), it would stand to reason that the authors would have attempted to show that cultures treated with Wnt5a + Gremlin show improved GSIS. If not, then that would suggest that Wnt5a/BMPi is involved in ins+ cell development but not maturation.

We agree that the experiment was preliminary and therefore we removed it from MS. Furthermore, we have not included GSIS of β -cell derived in presence of WNT5A alone or WNT5A with Gremlin1, showing some functional capacity (Fig. 7J). WNT5A with Gremlin1 treatment caused the significant increase in secreted insulin in response to high glucose compared to none treated cells. However, the GSIS of these cells do not reach the level observed for β -cell coculture with M-E20. Therefore, we conclude that the WNT5A regulates mostly β -cell development, and that β -cell maturation is regulated by cohort of factors that might include WNT5A and Gremlin.

Minor comments

Figure 2A and A’ seem to be redundant with A’ and can be moved to supplement.

Figures 2A-A’ have been moved to supplement as Fig. S3B-D. Instead, in the main panel of figures we included data for d3-14 with low/high glucose stimulation only (omitting KCl treatment, Fig. 1G).

Figure 3 is all transcript based so the descriptions need to reflect this. The title in the results suggests that growth factor and ECM proteins were analyzed.

The Fig. 3 title has been changed to “Human pancreatic M-E17 and M-E20 cells transcriptomes are jointly enriched in secreted factors that have their receptors present in hESC-derived EPs”

Reviewers' Comments:

Reviewer #1:

Remarks to the Author:

In the manuscript "Human pancreatic microenvironment promotes β -cell differentiation via non-canonical 2 WNT5A/JNK and BMP signaling", the authors Chmielowiec et al. describe a comprehensive analysis to identify several novel factors with a role in promoting β cell differentiation. Their methodology spans from identifying factors produced by primary M-E niche cells to incorporating and testing the effect of these factors in an improved protocol for the in vitro differentiation of hESC to β cells.

Wnt5A has been identified as an important component in this differentiation pathway, and as such, this report represents an important finding in the field of pancreatic development. Beyond this, the broader impact is less clear to me, as Wnt5A has already been shown to have an essential role in differentiating many cell types. Furthermore, evidence that Wnt/PCP pathway signalling has a vital role in β cell differentiation has already been published (Bader et al., 2016, Nature). However, a direct role for Wnt5A alone could not be validated (Vethe et al., 2019, Frontiers). Therefore, a thorough discussion of these discrepancies is required. For example, could the differences be explained because different starting progenitors were used compared to the submitted manuscript?

The general methodology used throughout is to culture PP or EP with primary M-E or factors that they produce. The authors test the effectiveness of M-E at different foetal stages, but the PP and EP are the same throughout. The authors need to explain this detail. Do these progenitor cells vary, so there may be an argument for matching the developmental stages/time point of EP/PP with the M-E? The fact that the authors repeat the experiments using different hESC lines goes some way to validate their findings, but it is unclear how this point is addressed.

It is good to see that the authors tested different methods for generating EP from hESC, but interesting, and not expanded upon, that the supposed most valid methods (i.e. 3D culture) gave the least improvement in results with WntA addition at the final step. Could the authors respond to this?

Wnt5A promoted β cell differentiation at higher (500ng/ml) but not lower concentrations – is there any evidence that this is physiologically relevant for a microenvironment?

Is INS/c-peptide production always a good proxy for glucose responsiveness, which the authors state is the actual physiological test for β cells? INS/c-peptide production measurement by IF etc., are used interchangeably, it seems at times with glucose responsiveness. (Or at least it is not always clear why one method is chosen over the other).

Specific points:

- 1) Line 108: M-E at Wk9.1, 10.6, 13, 14.6, 16.3, 17.5 and 20.1 are used. Given that the first wave is at Wk8-9, is there a possibility that the authors have missed important initial microenvironment factors?
- 2) Figure 1b: Why is the distribution of the endothelial markers not even and instead clustered in the centre?
- 3) Figure 1e, line 141: Co-culturing PP cells with M-E9 does not induce C-peptide/INS expression – but there is a wave of β cell production at this time point?
- 4) Figure 1f, line 170: How does this compare to β cells produced by other currently available methods? Good that it is compared to human islets.
- 5) Line 187, figures 2b and 2d: Why does figure 2b measure INS and 2d measure C peptide? Inconsistent, especially as 2c and e both measure C-peptide.
- 6) Line 190: Figure 1E and 2D are not directly comparable – should this be 1D and 2D?
- 7) Line 194: There is no evidence that these INS+ cells are also Glucose responsive.
- 8) Line 200: Missing a full stop.

- 9) Line 212: Figure 3C only shows Wnt, not Wnt5A.
- 10) Line 235: repetition of "using IF".
- 11) Line 237: Only one concentration is shown for LIF in Figures 4d and f.
- 12) Figure S7e, line 273: It is interesting here that the culture method that does not use a 3D organoid step is the one that produces the most significant increase in INS producing cells with Wnt5A treatment.
- 13) Figure S10C might be better in the main figures?
- 14) Line 351, Figure 6F: It would be helpful to specify that multiple bands likely represent different isoforms of JNK in the Western blot and state the antibodies used in the Methods section.
- 15) Line 470: error in "pathways".
- 16) Line 818: Font change.

Reviewer #3:

Remarks to the Author:

The authors have adequately addressed the reviewers suggestions and provided a thoughtful response. The manuscript is an important contribution to our understanding of supporting cell types in the developing human pancreas that participate in control of endocrine differentiation.

We thank the Reviewers for all their comments and overall positive assessment of our work. Following the suggestions, we have updated the MS text and Figures as indicated below. The detailed response to each comment is included below, in blue font. Thank you again for your time and input.

Reviewers' comments:

Reviewer #1 (Remarks to the Author):

In the manuscript "Human pancreatic microenvironment promotes β -cell differentiation via non-canonical 2 WNT5A/JNK and BMP signaling", the authors Chmielowiec et al. describe a comprehensive analysis to identify several novel factors with a role in promoting β cell differentiation. Their methodology spans from identifying factors produced by primary M-E niche cells to incorporating and testing the effect of these factors in an improved protocol for the *in vitro* differentiation of hESC to β cells.

Wnt5A has been identified as an important component in this differentiation pathway, and as such, this report represents an important finding in the field of pancreatic development. Beyond this, the broader impact is less clear to me, as Wnt5A has already been shown to have an essential role in differentiating many cell types. Furthermore, evidence that Wnt/PCP pathway signalling has a vital role in β cell differentiation has already been published (Bader et al., 2016, Nature). However, a direct role for Wnt5A alone could not be validated (Vethe et al., 2019, Frontiers). Therefore, a thorough discussion of these discrepancies is required. For example, could the differences be explained because different starting progenitors were used compared to the submitted manuscript?

We included Bader et al., work in our manuscript but thank you for pointing out that this was insufficiently discussed. Bader et al., 2016 and Vethe et al., 2019 both focused on the role of WNT/PCP in the β -cell maturation during early postnatal period, in adult human islets or at the final stage of *in vitro* β -cell differentiation. In contrast, we asked here a different question, namely whether WNT5A secreted by microenvironment controls β -cell fate commitment, defined as β -cell induction from pancreatic progenitors during development. Differentiation of β -cells *in vitro* follows stages of pancreatic *in vivo* development and every step is unique and requires precise analysis. In conclusion, our study uncovers the previously unstudied role of WNT5A signalling during human β -cell development.

We have now added to the discussion (lines 452-459):

“In the work of Bader et al.⁸⁷ WNT5A increased insulin secretion of human microislets *in vitro*, whereas Vethe et al.⁸⁸ applied WNT5A and B together on late stage (S7) β -cells and observed increased number of bi-hormonal INS+/GCG+ cells. In contrast, we show accelerated specification of EPs into functional, glucose-responsive (Figure 7J) and GCG negative (Figure S12) β -cells. The discrepancy might result from stage-specific response to WNT5A as we have tested its influence on β -cell fate acquisition from EPs, while the others in mature or maturing β -cells. Further, WNT5A might act through different mechanisms that we uncovered in our research, while they were not shown directly in former works.”

The general methodology used throughout is to culture PP or EP with primary M-E or factors that they produce. The authors test the effectiveness of M-E at different foetal stages, but the PP and EP are the same throughout. The authors need to explain this detail. Do these progenitor cells vary, so there may be an argument for matching the developmental stages/time point of EP/PP with the M-E? The fact that the authors repeat the experiments using different hESC lines goes some way to validate their findings, but it is unclear how this point is addressed.

Thank you for asking an interesting question. Yes, PPs, EPs and mesenchyme change over time. It was nicely demonstrated by various scRNA-Seq-based discoveries from various groups both on murine (including our work, Scavuzzo et al. Nat Com 2018) and human pancreatic cells (both foetal and stem cell derived). However, we do not know enough about differences between PPs or EPs from first-wave and second-wave or the mechanisms regulating these progenitors' further differentiation at different stages of development. For example, the early endocrine cell development might be independent of mesenchyme-derived signals. We have here used PPs/EPs at the same differentiation day, and it is indeed possible that they correspond to a certain embryonic day PPs/EPs *in vivo*, and thus might not be responsive to signals from M-E from other foetal stages. At the end of these *in vitro* stages, PPs/EPs are already matured and mostly synchronised to progress to the next developmental stage. Thus, it is possible that using less advanced PPs (e.g. day 8) with M-E9 or earlier would trigger first-wave like differentiation. Indeed, some prematurely induced (from NKX6-1 negative PPs) endocrine cells, which are bi-hormonal immature α -cells, arise earlier during PP stage in 3D protocols (see Melton lab papers - Veres et al., 2019 and a follow-up Peterson et al., 2020). However, the majority of endocrine cells are born during the second wave and we focused here on signals directing these events.

We have now added the following sentence to the first paragraph of the Discussion chapter (lines 420-422): "Inversely, using PPs differentiated for a shorter time (e.g. 8 days instead of 9 days) in coculture with M-E9 could induce first wave-like endocrine differentiation if it is triggered by M-E factors."

It is good to see that the authors tested different methods for generating EP from hESC, but interesting, and not expanded upon, that the supposed most valid methods (i.e. 3D culture) gave the least improvement in results with WntA addition at the final step. Could the authors respond to this?

Please see the response to point 12 below.

Wnt5A promoted β cell differentiation at higher (500ng/ml) but not lower concentrations – is there any evidence that this is physiologically relevant for a microenvironment?

Thank you for bringing up this interesting question. First, we would like to point out that WNT5A at lower concentration (100 ng/mL) was also effective (Fig. 4D, F; Fig. S6B-D), highlighted by statistical significance. However, at 500 ng/mL the effect on β -cell induction was stronger (Fig. 4D, F, S6D $p < 0.0001$; Fig. S6B $p = 0.0003$; Fig. S6C ns; post-hoc Dunnett multiple comparison test). Despite discussions with the other scientists and literature search, it is not easy to clearly state what is physiologically relevant WNT5A concentration during pancreatic β -cell development. We agree that 500 ng/mL is quite high concentration - compared to concentration of other growth factors commonly supplied as recombinant proteins in directed differentiation. Please note that both studies that you mentioned, Veathe et al., 2019, and Bader et al., 2016, used a similar amount of synthetic WNT5A - namely 400 ng/mL. Moreover, we have not tested any other concentrations than 100 and 500 and it

might be that 500 ng/mL might not be necessary for the induction of a full response during β -cell differentiation. Further, we applied recombinant WNT5A protein only once during a 3-day long treatment, and it is fair to assume a significant loss of the protein during this time.

Please also note, that we used WNT5A recombinant peptide produced in a hamster cell line and manufactured by Biotechne (formerly R&D Systems). This peptide is not a full length protein and although we contacted the manufacturer they can not attest whether all post-translational modifications are the same in this recombinant peptide as it is in native protein. WNT5A undergoes post-translational palmitoylation and glycosylation that are essential for WNT5A secretion and function. However, this WNT5A was tested positive for its biological effects and signalling transduction.

Is INS/c-peptide production always a good proxy for glucose responsiveness, which the authors state is the actual physiological test for β cells? INS/c-peptide production measurement by IF etc., are used interchangeably, it seems at times with glucose responsiveness. (Or at least it is not always clear why one method is chosen over the other).

We agree that INS or c-peptide synthesis does not reflect the glucose responsiveness and we never intended to say so. Only the GSIS analysis used in our study is a good indication of glucose-responsiveness of *in vitro* derived β -cells. We went carefully over the MS to make sure all results are clearly stated and not erroneously misinterpreted.

Specific points:

1) Line 108: M-E at Wk9.1, 10.6, 13, 14.6, 16.3, 17.5 and 20.1 are used. Given that the first wave is at Wk8-9, is there a possibility that the authors have missed important initial microenvironment factors?

Yes, it is possible. See also our response to point number 3 below. Moreover, we were not able to obtain human foetal pancreas at such early time points.

2) Figure 1b: Why is the distribution of the endothelial markers not even and instead clustered in the centre?

The endothelial cells were not spreading evenly as mesenchymal cells do. We also include here another example of staining with PECAM1 and DAPI.

M-E17h **PECAM1**/DAPI:

3) Figure 1e, line 141: Co-culturing PP cells with M-E9 does not induce C-peptide/INS expression – but there is a wave of β cell production at this time point?’

Thank you for bringing up this interesting point, which we have also addressed above (see page 2). The molecular mechanisms of so-called “first wave” pancreatic endocrine cells are unknown, thus might be different from the second wave, for instance independent of M-E cells. Furthermore we still know very little regarding what is the contribution of the endocrine cells born during first-wave to the adult islets. In mice, most of the “first wave” endocrine cells are polyhormonal and are lost during development. Moreover, please note that although we tried our best to recapitulate pancreatic microenvironment *in vitro*, we cannot exclude that during the M-E derivation and/or culture some subtypes of mesenchymal or endothelial cells are lost or their secretome is affected by culture condition.

4) Figure 1f, line 170: How does this compare to β cells produced by other currently available methods? Good that it is compared to human islets.

We have not directly compared insulin secretion upon M-E co-culture to other differentiation protocols, but rather human cadaveric islets. The reason is that we aimed to identify novel factors that would induce endocrine fate in PPs/EPs without additional factors usually added at corresponding differentiation stages. Therefore, we have not aimed here to create an improved protocol that should be directly compared with existing protocols. Also, as protocol using foetal-derived cells as part of coculture as in Fig. 1f would not be practical. Importantly, up to the point when coculture/select secreted growth factors were applied, we used various established protocols and obtained comparable efficiency of PP/EP differentiation based on IF and flow cytometry analysis of widely used markers. We are also convinced that comparison with other systems via data mining might be difficult as there are multiple factors affecting GSIS, and that indeed inclusion of human islets is informative.

5) Line 187, figures 2b and 2d: Why does figure 2b measure INS and 2d measure C peptide? Inconsistent, especially as 2c and e both measure C-peptide.

Thank you for bringing it to our attention. We erroneously labelled 2b, it now reads “C-PEP” instead of “INS”. We have also removed “and INS “ at line 188 of the manuscript.

6) Line 190: Figure 1E and 2D are not directly comparable – should this be 1D and 2D?

We aimed to compare Fig. 2C (ECM) and E (CM) to Fig. 1E. (e.g., 1.99- and 2.74-fold change in the number of C-PEP+ cells was seen for M-E20 CM and ECM, respectively, while 8.3-fold change for M-E20 co-culture). Therefore, we have updated the text accordingly.

Previously lines 189-190 read: “However, the efficiencies were lower than in the coculture experiments (**Figures 1E and 2D**)” and now they read: “However, the efficiencies were lower than in the coculture experiments (see **Figure 1E**).”

7) Line 194: There is no evidence that these INS+ cells are also Glucose responsive.

Indeed, we have not tested glucose-responsiveness upon ECM/CM treatment as we further focused on identification of individual factors secreted by M-E cells. We have performed GSIS analyses for PP/EP co-cultures and WNT5A+-GREM treatment.

8) Line 200: Missing a full stop.

Thank you for pointing this out, we have added the missing full stop.

9) Line 212: Figure 3C only shows Wnt, not Wnt5A.

Thank you for pointing this out, we have corrected “WNT5A” to “Wnt”.

10) Line 235: repetition of "using IF".

Thank you for your careful evaluation. We have changed “assessed by IF for the endocrine marker - CHGA, using IF (Figures 4C-D)” to “which we assessed by IF for the endocrine marker – CHGA (Figures 4C-D)”.

11) Line 237: Only one concentration is shown for LIF in Figures 4d and f.

Indeed, in the experiments performed in the H1 HUES cell line we assessed one concentration of our home-made LIF, at the concentration used for culture of mouse ESCs (~1 U/mL). However, we tested two concentrations of LIF for other HUES cell lines (Fig. S6). In these experiments we did not observe any difference between the two concentrations.

12) Figure S7e, line 273: It is interesting here that the culture method that does not use a 3D organoid step is the one that produces the most significant increase in INS producing cells with Wnt5A treatment.

In Figure S7E we applied WNT5A quite early (days 14 and 18) and observed its impact, namely INS+ cell induction, several days (days 18 and 22) before Pagliuca et al., did, who evaluated β -cell induction at day 27+. This suggests shortening the time necessary for INS+ cell *in vitro* differentiation. Importantly, in these experiments WNT5A was applied without any other growth factors applied at these stages. Please also note that when fold changes of UT vs WNT5A-treated conditions are compared, differences between 2D and 3D protocols are comparable within simultaneously conducted experiments (Fig. S7E).

When we applied WNT5A later (Figure S9A-B) - at the EN (endocrine progenitor) stage of the 3D Pagliuca protocol (day 20), and in the presence of other differentiation factors - we observed 27% to 46% increase of INS+ cell number at day 24, which is more comparable to effects observed in 2D protocols and to efficiencies obtained by Pagliuca et al., at day 27+. Please also note results in 3D co-culture experiments of EP stage spheroids (day 20) with WNT5A-KO M-E cells (Figure 5G), which shows that M-E cells deprived of WNT5A lose their ability to induce INS+ cells.

Together, we are convinced that the WNT5A effects in 3D cultures are not less significant than in 2D culture experiments.

13) Figure S10C might be better in the main figures?

Thank you for your suggestion. We have moved results pointing to Fzd3 as the receptor of WNT5A in pancreatic EPs from Fig. S10B and C to the main Figure 6A and B, respectively. We have accordingly updated references to figures in the Results section of the manuscript, and updated figure legends.

14) Line 351, Figure 6F: It would be helpful to specify that multiple bands likely represent different isoforms of JNK in the Western blot and state the antibodies used in the Methods section.

We have added the following sentence to the Figure 6 legend: “Multiple bands of p-JNK and JNK correspond to 54 and 46 kDa isoforms.”

All antibodies are listed in table S2 and in the Reporting summary. These particular antibodies are manufactured by Cell Signalling (#4668 and #9252), and widely cited. Two JNK isoforms, p54 and p46, were detected as also suggested on antibody datasheet. We have also updated labelling in Fig. 6.

15) Line 470: error in "pathways".

Thank you. We have corrected it.

16) Line 818: Font change.

Thank you, again. We have corrected the font.